# Discovering Spoofing Attempts on Language Model Watermarks

**Thibaud Gloaguen** [1]   **Nikola Jovanović** [1]   **Robin Staab** [1]   **Martin Vechev** [1]

## Abstract

LLM watermarks stand out as a promising way to attribute ownership of LLM-generated text. One threat to watermark credibility comes from spoofing attacks, where an unauthorized third party forges the watermark, enabling it to falsely attribute arbitrary texts to a particular LLM. Despite recent work demonstrating that state-of-the-art schemes are, in fact, vulnerable to spoofing, no prior work has focused on post-hoc methods to discover spoofing attempts. In this work, we for the first time propose a reliable statistical method to distinguish spoofed from genuinely watermarked text, suggesting that current spoofing attacks are less effective than previously thought. In particular, we show that regardless of their underlying approach, all current learning-based spoofing methods consistently leave observable artifacts in spoofed texts, indicative of watermark forgery. We build upon these findings to propose rigorous statistical tests that reliably reveal the presence of such artifacts and thus demonstrate that a watermark has been spoofed. Our experimental evaluation shows high test power across all learning-based spoofing methods, providing insights into their fundamental limitations and suggesting a way to mitigate this threat. We make all our code available here.

## 1. Introduction

The abilities of large language models (LLMs) to generate human-like text at scale (Bubeck et al., 2023; Dubey et al., 2024) come with a growing risk of potential misuse. This makes reliable detection of machine-generated text increasingly important. Researchers have proposed the concept of watermarking: augmenting generated text with an imperceptible signal that can later be detected to attribute ownership of a text to a specific LLM (Kirchenbauer et al., 2023; Kudi-

tipudi et al., 2024; Christ et al., 2024b). As such watermarks are actively deployed on top of consumer LLMs (Dathathri et al., 2024a) and widely embraced by regulators (Biden, 2023; CEU, 2024), ensuring their reliability is crucial.

**LLM watermarks**   To embed a signal, at each step of generation, using a private key $\xi$, the watermark algorithm *scores* each token, preferentially sampling higher-scoring ones. While a wide range of watermarking schemes have been proposed (Christ et al., 2024b; Kuditipudi et al., 2024; Aaronson, 2023), the most studied and at this time the only ones deployed in prominent consumer LLMs (Dathathri et al., 2024a) are from the *Red-Green* (Kirchenbauer et al., 2023) family. In Red-Green watermarks, the algorithm uses $\xi$ and a few previous tokens (*context*) to partition the vocabulary into *green* and *red* tokens. It then increases the probability of sampling green tokens. Given a text, the *watermark detector* first computes the *color* of each token under $\xi$, wherein a high proportion of green tokens in this *color sequence* indicates watermarked text.

**Spoofing attacks**   Recent works have demonstrated targeted attacks on Red-Green watermarks that allow for impersonating (*spoofing*) the watermark (Sadasivan et al., 2023; Jovanović et al., 2024; Gu et al., 2024; Zhang et al., 2024). In spoofing attacks, a malicious actor (*spoofer*) generates, without knowing the private key $\xi$, a text that is detected as watermarked. State-of-the-art attacks are *learning-based* and adhere to a common pipeline (see App. J for a full taxonomy). First, the malicious actor queries the targeted model to build a dataset $\mathcal{D}$ of genuinely watermarked text. Then, by either applying statistical methods (Jovanović et al., 2024), integer programming (Zhang et al., 2024), or fine-tuning (Gu et al., 2024), the spoofer learns how to forge the watermark and can generate watermarked text without additional queries to the original model (Step 1 in Figure 1).

Being able to generate spoofed text at scale poses a serious threat to the credibility of watermarks. Spoofed text can be falsely attributed to the model provider, causing reputational damage, or used as an argument to evade accountability (Zhou et al., 2024). Moreover, in the case of multi-bit watermarks that embed client IDs in generated text (Wang et al., 2024), spoofing attacks can be used to impersonate and incriminate a specific user.

---

[1]ETH Zurich. Correspondence to: Thibaud Gloaguen <tgloaguen@student.ethz.ch>.

*Proceedings of the 42ⁿᵈ International Conference on Machine Learning*, Vancouver, Canada. PMLR 267, 2025. Copyright 2025 by the author(s).

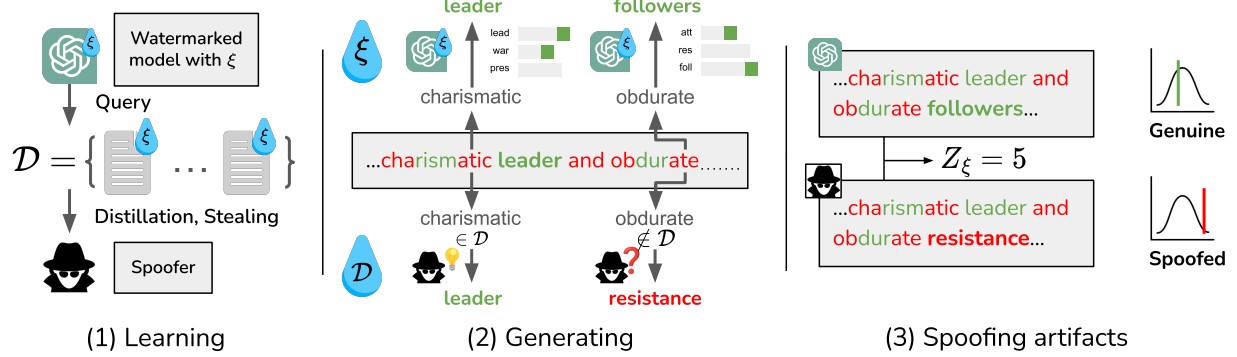

Figure 1: Overview of why spoofed text contains measurable artifacts. First, in (1), the spoofer generates a dataset $\mathcal{D}$ of $\xi$-watermarked texts from which they learn the watermark. As (2) illustrates, when later generating text, the spoofer is better at sampling a green token if (and only if) the context and the sampled token were in $\mathcal{D}$. This uncertainty introduces artifacts in the spoofed text. In contrast, the genuine watermarking algorithm is consistent with respect to the context and hence contains no such artifacts. Lastly, in (3), we build statistical tests for discovery of these artifacts, distinguishing between spoofed and $\xi$-watermarked texts even if their Z-scores $Z_\xi$ computed using the watermark detector are the same.

**Discovering spoofing attempts** In this work, we show for the first time that state-of-the-art spoofing attacks leave artifacts in the generated text that can be used to distinguish between spoofed text and text generated with knowledge of the private key (Step 2 in Figure 1). This suggests that, unlike previously thought, *simply fooling the watermark detector is not enough to generate text that is indistinguishable from genuine watermarked text*. The high-level intuition behind these artifacts is that, at each step of generation, a spoofer has a chance to emit a green token only if the context and that token are present in their training data $\mathcal{D}$, previously obtained by querying the watermarked model. If the context is not in $\mathcal{D}$, the spoofer is forced to select the next token independently of its color. Leveraging these artifacts, we construct statistical tests that can effectively distinguish between spoofed text and genuine watermarked text generated with the private key (Step 3 in Figure 1).

In addition to enabling the discovery of spoofing attempts on widely researched, deployed, and attacked Red-Green watermarking schemes, we show in App. G that our tests generalize to all schemes that are vulnerable to learning-based spoofing (Aaronson, 2023; Kuditipudi et al., 2024).

**Key contributions** Our main contributions are:

- We provide the first in-depth analysis of artifacts in spoofed text, highlighting common limitations of learning-based watermark spoofing methods (§3).

- We design rigorous statistical tests to practically distinguish spoofed and genuine watermarked texts (§4).

- We provide extensive validation of our test hypotheses and empirically show that our tests achieve arbitrarily high power given a long enough text (§5).

## 2. Background and Related Work

In this section we introduce the necessary background on LLM watermarking, and discuss related work.

**LLM watermarks** Given a sequence of tokens (*text*) from a vocabulary $\Sigma$, an autoregressive language model (LM) $\mathcal{M}$ outputs a logit vector $l$ of unnormalized next-token probabilities, used to sample the next token. *LM watermarking* is a process of embedding a signal within the generated text $\omega$ using a private key $\xi$, such that this signal is later detectable by any party with access to $\xi$ using a watermark detector $D_\xi$. We set $D_\xi(\omega) = 1$ if the signal is detected. We call a text $\omega$ generated by the watermarking algorithm a $\xi$-*watermarked* text, and a text where $D_\xi = 1$ a *watermarked* text.

The most prominent approach to LLM watermarking are Red-Green watermarks (Kirchenbauer et al., 2024; Zhao et al., 2024; Lee et al., 2024; Wu et al., 2024; Yoo et al., 2024; Fernandez et al., 2023; Liu et al., 2023; Fairoze et al., 2023; Ren et al., 2024; Lu et al., 2024; Guan et al., 2024; Zhou et al., 2024; Dathathri et al., 2024a), which all share a common structure. Let $\omega_t \in \Sigma$ be the token generated by the LM at step $t$, $h \in \mathbb{N}$ the watermark's *context size* (we refer to $h$ previous tokens $\omega_{t-h:t-1}$ as the *context*), $\xi \in \mathbb{N}$ the watermark's private key, $H : \Sigma^h \to \mathbb{N}$ a hash function, $PRF : \mathbb{N} \times \mathbb{N} \to \mathcal{P}(\Sigma)$ a pseudorandom function, and $\gamma, \delta \in \mathbb{R}$ watermark parameters. At each step $t$, $PRF$ uses the hash of the context $H(\omega_{t-h:t-1})$ and the private key $\xi$ to partition the vocabulary $\Sigma$ into two *colors*, $\gamma|\Sigma|$ *green* tokens (*greenlist*) and the remaining *red* tokens (*redlist*), where $\gamma$ is the watermark parameter. To insert the watermark, we modify the logit vector $l_t$ by increasing the logit of each green token by $\delta > 0$. While many hash functions $H$ have been proposed (Kirchenbauer et al., 2024), we fo-

cus on two variants proposed in Kirchenbauer et al. (2023): *SumHash* and *SelfHash*. The shift by $\delta$ increases the ratio of green tokens in generated text, which is detectable by the detector. Namely, given a text $\omega \in \Sigma^T$, the watermark detector $D_\xi$ determines the number of green tokens $n_{green}$ and computes $Z_\xi(\omega) = (n_{green} - \gamma T)/\sqrt{T\gamma(1-\gamma)}$, which under the null hypothesis follows a standard normal distribution. Finally, $D_\xi(\omega) = 1$ (i.e., $\omega$ is considered watermarked) if $Z_\xi(\omega) > \rho$. As in Kirchenbauer et al. (2023), we set $\rho = 4$.

Other alternative approaches to LLM watermarks are proposed by, among else, Christ et al. (2024b); Kuditipudi et al. (2024); Hu et al. (2024); Aaronson (2023). Among these, prior work demonstrates learning-based spoofing attacks on Kuditipudi et al. (2024) and Aaronson (2023) (see App. G).

**LLM watermark spoofing**  A threat to watermark credibility are spoofing attacks, as they can lead to falsely attributing text ownership to a model provider. One type of spoofing attack is *piggyback* spoofing (Pang et al., 2024), where an attacker substitutes a few tokens in a genuinely watermarked text to produce a spoofed text, simply leveraging the robustness of the watermarking scheme. Another type of spoofing attacks is *step-by-step* spoofing (Pang et al., 2024; Zhou et al., 2024; Wu & Chandrasekaran, 2024), where for every spoofed text, the attacker queries the watermarked model at each step of its generation process. Lastly, state-of-the-art spoofing attacks are *learning-based* spoofing attacks (Jovanović et al., 2024; Gu et al., 2024; Zhang et al., 2024), where an attacker first queries the watermarked model to build a watermarked dataset $\mathcal{D}$ and then learns the watermark from such a dataset. Unlike the two other spoofing techniques, learning-based spoofers are able to produce arbitrary watermarked text at a low cost without relying on the attacked model during generation. We extend the discussion on watermark spoofing in App. J.

Among learning-based spoofers, there are two approaches that generalize across most Red-Green schemes: *Stealing* (Jovanović et al., 2024) and sampling-based *Distillation* (Gu et al., 2024). Stealing approximately infers the vocabulary splits by comparing the frequencies of tokens in $\mathcal{D}$ (conditioned on the same context) with human-generated text, and uses this information to generate spoofed text using an auxiliary LM. In contrast, Distillation directly fine-tunes an auxiliary LM on $\mathcal{D}$, effectively distilling the watermark into the model's weights. Both Stealing and Distillation are applicable on Red-Green schemes, but Distillation also expands to other schemes (see App. G).

**Spoofing defenses**  Some watermarking schemes have been specifically designed for attribution and hence are more resistant to spoofing (Zhou et al., 2024; Christ et al., 2024b; Christ & Gunn, 2024; Fairoze et al., 2023). Yet, in their focus on providing attribution, they trade off other desirable watermark properties (Jovanović et al., 2024; Kirchenbauer et al., 2024), preventing their practical adoption (Dathathri et al., 2024b). Similarly, even in Red-Green schemes, higher values of the context length $h$ reduce the success rate of spoofing at the cost of watermark robustness.

**Broader work on LLM watermarking**  Other directions in the realm of LLM watermarking includes scrubbing attacks (Jovanović et al., 2024; Wu & Chandrasekaran, 2024; Chang et al., 2024), detection of the presence of a watermark (Tang et al., 2023; Gloaguen et al., 2025; Liu et al., 2025; Gao et al., 2025), and attempts to imprint the watermark into the model weights (Li et al., 2024; Christ et al., 2024a).

# 3. Can Spoofing Attempts Be Discovered?

In this section, we discuss the discoverability of spoofing, introduce the problem of distinguishing $\xi$-watermarked and spoofed texts, and formalize it within a hypothesis testing framework (§3.1). We describe the intuition behind our approach (§3.2), that we later present in detail in §4.

## 3.1. Problem statement

Current spoofing methods (*spoofers*) are typically evaluated based on their success rate in generating high-quality watermarked text. Yet, due to the limitation of learning from a finite dataset of watermarked text, we hypothesize that these spoofers, despite adopting fundamentally different approaches, may all leave similar artifacts in spoofed texts.

Showing the existence of such artifacts would give valuable insight into the shared limitations of current state-of-the-art watermark spoofers. Moreover, reliably identifying them would enable us to distinguish between $\xi$-watermarked and spoofed texts, lowering the effective accuracy of spoofers, without compromising other desirable properties, as is often the case when trying to specifically design watermarking schemes more resistant to spoofing (see §2).

Concretely, we assume the perspective of the *model provider* with a private key $\xi$ and a model $\mathcal{M}$. We receive a text $\omega \in \Sigma^T$ that is flagged as watermarked by our detector $D_\xi$, and aim to decide whether it was generated using our private key $\xi$, or by a spoofing method. Our threat model also includes the case where we receive a *set of texts* from the same source, whose concatenation we denote as $\omega \in \Sigma^T$ for simplicity (see the bottom of §4.2 for details). We assume that our private key $\xi$ was not simply leaked; else, spoofed texts are hardly distinguishable from $\xi$-watermarked texts.

**Formalization**  Determining whether a text $\omega$ was spoofed can be formulated within the hypothesis testing framework:

$$H_0 : \text{The text } \omega \text{ is } \xi\text{-watermarked,}$$
$$H_1 : \text{The text } \omega \text{ is spoofed.} \tag{1}$$

We introduce the random variable $\Omega \in \Sigma^T$ and the received text $\omega \in \Sigma^T$ is a realization of $\Omega$. We note that the distribution of $\Omega$ under the null hypothesis and its distribution under the alternative hypothesis are different. Similarly, let $X \in \{0, 1\}^T$ be the associated sequence of (non-i.i.d.) Bernoulli random variables, where $X_t = 1$ represents the event where the token $t$ is green, and let $x \in \{0, 1\}^T$ be the observed color of $\omega$ under $D_\xi$ (realization of $X$). In this hypothesis testing framework, the challenge is to build a statistic $S(\Omega)$ that satisfies two key properties. First, the distribution of $S(\Omega)$ under the null hypothesis should be known in order to rigorously control the Type 1 error. Second, the distributions of $S(\Omega)$ under the null and $S(\Omega)$ under the alternative should be different, enabling us to distinguish spoofed and $\xi$-watermarked texts.

### 3.2. Artifact: dependence between the color sequence and the context

Next, we explain why spoofed texts contain observable artifacts, as was illustrated in Figure 1.

**A simple example** To expand on this intuition, we start by considering an example of a perfect spoofer that produced the text $\omega \in \Sigma^T$, and knows the color of a token $\omega_t$, if and only if $\omega_{t-h:t} \in \mathcal{D}$, where $\mathcal{D}$ is the training data of the spoofer. Otherwise, if $\omega_{t-h:t} \notin \mathcal{D}$, we assume that the spoofer has chosen $\omega_t$ independently of its color. Let $I_\mathcal{D} : \Sigma^{h+1} \to \{0, 1\}$ be the indicator function of the presence of a $(h+1)$-gram in $\mathcal{D}$. $I_\mathcal{D}$ can be interpreted as the knowledge the spoofer has over the vocabulary splits. From above, we can assume that for all $t \in \{h+1, \dots, T\}$:

$$P(X_t = 1 | I_\mathcal{D}(\Omega_{t-h:t}) = 1) \geq P(X_t = 1 | I_\mathcal{D}(\Omega_{t-h:t}) = 0)$$
if the text is spoofed; $\qquad$ (2a)
$$P(X_t = 1 | I_\mathcal{D}(\Omega_{t-h:t}) = 1) = P(X_t = 1 | I_\mathcal{D}(\Omega_{t-h:t}) = 0)$$
if the text is $\xi$-watermarked. $\qquad$ (2b)

Equations (2a) and (2b) reflect that the knowledge of the vocabulary split at token $t$ helps the spoofer to color $\omega_t$ green, which is its original goal. For a $\xi$-watermarked text, the knowledge of a potential spoofer has no influence on its coloring. Hence, we may be able to use $I_\mathcal{D}$ to distinguish whether a sentence is spoofed or not. We now generalize this intuition to more realistic spoofing scenarios.

**Color sequence depends on the context distribution** In practice, learning how to spoof may require observing an $(h+1)$-gram multiple times. Moreover, spoofing techniques may, albeit not necessarily explicitly, have different levels of certainty regarding the color of a token given a context. Therefore, we generalize $I_\mathcal{D} : \Sigma^{h+1} \to [0, 1]$ to be the function of the frequencies of $(h+1)$-grams in $\mathcal{D}$. We make a natural assumption that the higher the frequency of $\omega_{t-h:t}$

in $\mathcal{D}$, the more certain a spoofer is regarding the color of the token $\omega_t$. For now, we will also assume that for each token in $\xi$-watermarked text, $I_\mathcal{D}$ is independent of its observed color. For $\forall t \in \{h+1, \dots, T\}$, we assume:

$X_t$ is not independent from $I_\mathcal{D}(\Omega_{t-h:t})$
if the text is spoofed; $\qquad$ (3a)
$X_t$ is independent from $I_\mathcal{D}(\Omega_{t-h:t})$
if the text is $\xi$-watermarked. $\qquad$ (3b)

This dependence between the color and $I_\mathcal{D}(\Omega_{t-h:t})$ results in spoofing artifacts under the alternative.

**Influence of the LM** Counterintuitively, the independence assumed in Eq. (3b) may be violated. To generate $\omega_t$, the model provider first computes the logit vector $l_t$ knowing $\omega_{<t}$. Then, it computes the greenlist defined by $PRF(H(\omega_{t-h:t-1}), \xi)$, and increases the logits of green tokens by $\delta$. Finally, it samples from the newly defined probability distribution to generate the token $\omega_t$. The greenlist itself is thus indeed independent of $I_\mathcal{D}(\Omega_{t-h:t})$. Yet, $l_t$ was originally computed using $\omega_{<t}$ due to the autoregressive property of the model $\mathcal{M}$, and hence may not be independent of $I_\mathcal{D}(\Omega_{t-h:t})$.

To illustrate this point, consider a case where the token $w_t$ is the only viable continuation of $\omega_{t-h:t-1}$, i.e., $l_t$ is low-entropy. Then, Bayes' theorem implies that $I_\mathcal{D}(\omega_{t-h:t})$ is likely to be high. On the other hand, the logit increase of $\delta$ has less influence on the sampling, as it is less likely to cause a token other than $w_t$ to be sampled—thus, the color of $w_t$ is effectively random, i.e., $P(X_t = 1) \approx \gamma$, even for $\xi$-watermarked text. Hence, the events $P(X_t = 1) \approx \gamma$ and $I_\mathcal{D}(\Omega_{t-h:t})$ *is high*, are correlated, as they occur simultaneously in case of low entropy. We investigate this dependence pattern and confirm it experimentally in App. C.

With this in mind, to properly control for Type 1 error, we need to design a test statistic $S$ where this dependence pattern is known or can be learned for $\xi$-watermarked texts. Moreover, to maintain power, we aim to distinguish this dependence from the dependence present in the case of spoofed text, as described above in Equations (3a) and (3b).

## 4. Designing a Test Statistic

We proceed to introduce our test statistic $S$, deriving fundamental results regarding its distribution under the independence assumption from Eq. (3b), and in the more general case where it may be violated (§4.1). Then, we present and discuss two concrete instantiations of $S$ (§4.2).

### 4.1. Controlling the distribution

We introduce the main results regarding the distribution of $S(\Omega)$ under the null hypothesis.

**Color-score correlation**    Let $\omega \in \Sigma^T$, sampled from $\Omega$, denote the text of length $T$ received by the model provider, $x \in \{0,1\}^T$, sampled from $X$, denote its color sequence under $D_\xi$, and $y \in [0,1]^T$ denote a sequence of *scores* for each token sampled from a sequence of $T$ random variables $Y$. We defer the construction of $Y$ to §4.2, where we will build on the intuition from §3.2. As the test statistic, we use the sample Pearson correlation coefficient between $x$ and $y$,

$$S(\omega) = \frac{\sum_{t=1}^T (x_t - \bar{x})(y_t - \bar{y})}{\sqrt{\sum_{t=1}^T (x_t - \bar{x})^2 \sum_{t=1}^T (y_t - \bar{y})^2}}. \quad (4)$$

**Independence case**    We first study the distribution of $S(\Omega)$ under the assumption that $X_i$ and $Y_i$ are independent for all $i$, as in Eq. (3b) (we refer to this as *cross-independence* between $X$ and $Y$). From this assumption, we derive:

**Lemma 4.1.** *Under the cross-independence between $X$ and $Y$, and technical assumptions (detailed in App. I), we have the convergence in distribution*

$$Z_S(\Omega) := \sqrt{T}S(\Omega) \xrightarrow{d} \mathcal{N}(0,1).$$

We defer the proof to App. I. Therefore, given a text $\omega$, we can compute a p-value using a two-sided Z-test on the statistic $Z_S(\omega)$, which is sampled from a standard normal distribution. We will refer to this test as the *Standard* method.

**The general case**    In practice, however, the assumption of cross-independence between $X$ and $Y$ does not always hold (see §3.2). We make a modeling assumption motivated by the results from the independent case. Let $\mu_\Omega := \mathbb{E}[S(\Omega)]$. Under the null hypothesis (and the practical considerations outlined below), we assume that

$$\sqrt{T}S(\Omega) \sim \mathcal{N}(\mu_\Omega, 1). \quad (5)$$

Compared to Lemma 4.1, the difference is that the normal distribution is offset by $\mu_\Omega$. This introduces a key challenge: finding a way to estimate $\mu_\Omega$. To this end, we propose to use $\omega_{\leq c}$, a prefix of $\omega$ of length $c$, to prompt our model $\mathcal{M}$ to generate a new sequence $\omega'$ of length $T' := T - c$ (which is a realization of $\Omega'$). In practice, we set $c = 25$. Given the shared prefix, we expect that $\Omega_{>c} \sim \Omega'$ and hence that $\mathbb{E}[S(\Omega_{>c})] = \mathbb{E}[S(\Omega')] = \mu_\Omega$. Then we introduce the statistic $Z_R(\Omega, \Omega')$, defined by

$$Z_R(\omega, \omega') = \frac{S(\omega_{>c}) - S(\omega')}{\sqrt{1/(T-c) + 1/T'}}. \quad (6)$$

Under the null hypothesis, we have that $Z_R(\Omega, \Omega') \sim \mathcal{N}(0,1)$, as $S(\omega_{>c})$ and $S(\omega')$ are two independent samples from a normal distribution. Therefore, in the general case, at the cost of higher computational complexity (since

we need to use the model to generate the new text), we can, as in the independent case, compute a p-value using a Z-test on the statistic $Z_R(\omega, \omega')$, which is sampled from a standard normal distribution. We later refer to this test as the *Reprompting* method. For consistency, in Reprompting experiments in §5, we use $T$ to implicitly refer to $T - c$. We additionally study Reprompting in greater details in App. F.

### 4.2. Concrete instantiations

In this section we instantiate the score sequence $Y$ and propose practical modifications to $S$.

**Construction of the token score**    We propose two instantiations of the score function $Y$: one that closely follows the intuition from §3.2, and another that aims to achieve the independence assumption from Lemma 4.1. Achieving cross-independence allows the construction of a test that does not require reprompting the model, hence reducing computational complexity.

**(h+1)-gram score**    For the first instantiation, the idea is to directly approximate $I_\mathcal{D}$, the function of $(h + 1)$-grams frequencies in $\mathcal{D}$. As $\mathcal{D}$ is not known to the model provider, we approximate it with a text corpus $\tilde{\mathcal{D}}$. We define

$$y_t := I_{\tilde{\mathcal{D}}}(\omega_{t-h:t}). \quad (7)$$

In practice, we set $\tilde{\mathcal{D}}$ to C4 (Raffel et al., 2020). We study the influence of $\tilde{\mathcal{D}}$ in App. D. Finally, to reduce the required size of $\tilde{\mathcal{D}}$ needed to obtain a good estimate of $I_\mathcal{D}$, we compute the frequency of unordered $(h + 1)$-grams. Because the independence assumption from Lemma 4.1 is not met here (see §5.1), we use Reprompting with this score.

**Unigram score**    For the second instantiation, the intuition is to trade-off between cross-independence and reflecting $I_\mathcal{D}$. Let $f : \Sigma \to [0,1]$ be the unigram frequency in human generated text. We define

$$y_t := f(\omega_{t-h}). \quad (8)$$

We look at the unigram frequency furthest from $t$, to make the dependence between $X$ and $Y$ negligible. Yet, we remain within the context window so $y_t$ partially reflects the information from $I_\mathcal{D}(\omega_{t-h:t})$ and hence still allows distinguishing spoofed and $\xi$-watermarked texts. We will see in §5.1 that cross-independence is satisfied for SumHash $h = 3$. Hence, in settings where cross-independence is verified, we use this score with the Standard method.

**Practical considerations**    In practice, we add modifications to the statistic $S$. First, as suggested in Kirchenbauer et al. (2023), we ignore repeated $h$-grams in the sequence $\omega$. This is required to enforce the independence assumption

within $X$ and the independence within $Y$. Second, to limit the influence of outliers on the score, we use the Spearman rank correlation instead of the Pearson correlation and further apply a Fisher transformation. This means that in Eq. (19), $x$ and $y$ are respectively replaced by $R(x)$ and $R(y)$, where $R$ is the rank function. Hence, the statistic $S(\omega)$ used in practice is defined as the arctanh of

$$\frac{\sum\limits_{t=1}^{T}(R(x)_t - \overline{R(x)})(R(y)_t - \overline{R(y)})}{\sqrt{\sum\limits_{t=1}^{T}(R(x)_t - \overline{R(x)})^2 \sum\limits_{t=1}^{T}(R(y)_t - \overline{R(y)})^2}}. \quad (9)$$

Therefore, we also use the variance $\sqrt{\frac{1.06}{T-3}}$ instead of $\sqrt{\frac{1}{T}}$ to reflect the influence of the rank function, as suggested in Fieller et al. (1957) for the i.i.d. case.

**Combining texts**  Given a set of texts from a single source, we concatenate all its elements to create a single text of size $T$. In particular, let $n \in \mathbb{N}$ and $\omega^1, \cdots, \omega^n \in \Sigma^{T_1} \times \cdots \times \Sigma^{T_n}$ such that $T_1 + \cdots + T_n = T$ for a given $T$. For the Standard method, we set $\omega := \omega^1 \circ \cdots \circ \omega^n$. For the Reprompting method, we compute $\omega'^1, \cdots, \omega'^n$ independently enforcing $T_i' = T_i - c$ and then set $\omega' := \omega_1' \circ \cdots \circ \omega_n'$ and define $\omega_{>c} := \omega_{>c}^1 \circ \cdots \circ \omega_{>c}^n$. We verify experimentally in App. B that the concatenation operation has no influence on the distribution of the statistic. Our experiments with large $T$ in §5 are thus conducted on concatenated texts.

**Extending to other watermarking schemes**  The framework can naturally be extended to most other watermarking schemes. On a high level, a watermarking scheme is a sequence of random vectors $\zeta_t$ and a mapping $w : \mathbb{R}^\Sigma \times \mathbb{R}^\Sigma \to \mathbb{R}^\Sigma$ such that the next token is sampled according to the logit vector $w(l_t, \zeta_t)$ instead of $l_t$. In the case of Red-Green watermark, $\zeta_t$ is simply the coloring of the vocabulary, and we set $x_t = \zeta_t[\omega_t]$ in Eq. (4). Hence, for any other schemes, when setting $x_t = \zeta_t[\omega_t]$, the results from §4.1 still hold. We show concrete instantiations of $x$ and experiment evaluation of our method for both AAR (Aaronson, 2023) and KTH (Kuditipudi et al., 2024) in App. G.

# 5. Experimental Evaluation

We present the results of our experimental evaluation. In §5.1, we validate the normality assumptions from §4.1. In §5.2, we validate the control of Type 1 error and evaluate the power of the tests from §4 on both spoofing techniques introduced in §2: *Stealing* (Jovanović et al., 2024) and *Distillation* (Gu et al., 2024). In App. A, we compare the test results across a wider range of spoofer LMs, and we show additional results with a different watermarked model $\mathcal{M}$, parameter combinations, and another prompt dataset. In

App. G, we show that the tests generalize to two additional watermarking schemes, AAR (Aaronson, 2023) and KTH (Kuditipudi et al., 2024), which only Distillation can spoof.

**Experimental setup**  We primarily focus on the KGW SumHash scheme, using a context size $h \in \{1, 2, 3\}$ and $\gamma = 0.25$. For $h \in \{1, 2\}$, we set $\delta = 2$. For $h = 3$, we use $\delta = 4$ for Stealing to ensure high spoofing rates and note that Distillation is unable to reliably spoof in this setting, and therefore is excluded from our $h = 3$ experiments. In each experiment, we generate either spoofed or $\xi$-watermarked continuations of prompts sampled from the news-like C4 dataset (Raffel et al., 2020), following the methodology from prior work of Kirchenbauer et al. (2023). For each parameter combination, we generate 10,000 continuations, each being between 50 and 400 tokens long. Then, we concatenate continuations (see §4.2) to reach the targeted token length $T$. Finally, each concatenated continuation is filtered by the watermark detector, and only watermarked sequences are kept. We use those concatenated continuations to compute the test statistic $S$. In practice, we have on average a total of $10^6/T$ samples per parameter combination.

We match the experimental setup from Jovanović et al. (2024) and Gu et al. (2024). In particular, we use LLAMA2-7B as the watermarked model. More specifically, in line with their original setups, we use the instruction fine-tuned version for Stealing and the completion version for Distillation. For the spoofer LM, we use MISTRAL-7B as the attacker for Stealing and PYTHIA-1.4B as the attacker for Distillation. Finally, for the spoofer training data $\mathcal{D}$, we use $\xi$-watermarked completions of C4. For Stealing, $\mathcal{D}$ is composed of 30,000 samples, each 800 tokens long, whereas for Distillation, $\mathcal{D}$ is composed of 640,000 samples, each 256 tokens long. We further study the impact of $|\mathcal{D}|$ in App. E.

## 5.1. Validating the normality assumption

In §4 we discuss two cases, each relying on one fundamental assumption. The *Independence case*: we assume independence between the color sequence $X$ and scores $Y$, from which we derive the normality of $S(\Omega)$ with a known mean (Lemma 4.1). For this case, we use the Standard method with the unigram score (Eq. (8)). The *General case*: we alternatively assume that $S(\Omega)$ is normally distributed with an unknown mean (Eq. (5)). Here, we use the Reprompting method with the $(h + 1)$-gram score (Eq. (7)). In Figure 2, we test the Independence case assumption by validating if $Z_S(\Omega)$ with the Standard method and unigram score follows a standard normal distribution (Top), and the General case assumption by validating the same for $Z_R(\Omega, \Omega')$ with the Reprompting method and $(h + 1)$-gram score (Bottom). We additionally perform both a Kolmogorov-Smirnov test for standard normality and a Pearson's normality test (not necessarily standard normal).

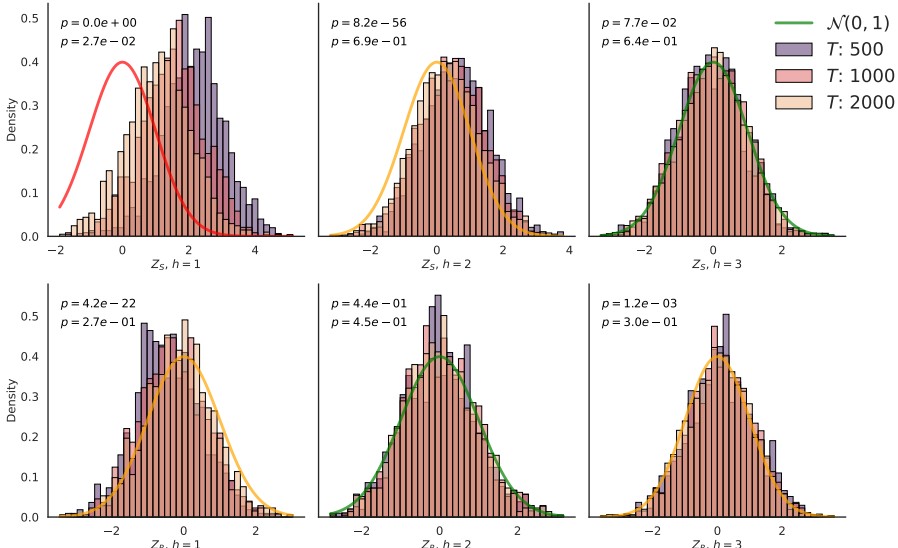

Figure 2: Histograms of $Z_S(\Omega)$ (*top*) and $Z_R(\Omega, \Omega')$ (*bottom*), with y-axes scaled to represent normalized density. The top row is computed using the unigram score and the Standard method, and the second row is computed using the $(h+1)$-gram score and the Reprompting method. A green line indicates that the $\mathcal{N}(0,1)$ hypothesis is not rejected (top p-value), an orange line that a normality test is not rejected (bottom p-value), and a red line that both are rejected at 5%.

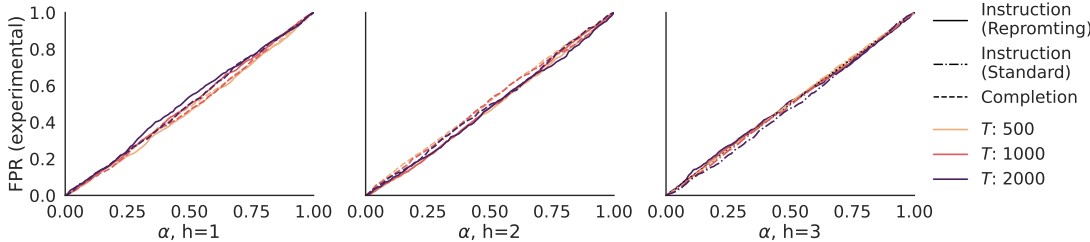

Figure 3: Experimental rejection rate of $\xi$-watermarked text on LLAMA2 7B.

Regarding the Independence case, we see that in the top row, $Z_S(\Omega)$ follows a standard normal distribution only for $h = 3$. This confirms our intuition behind the unigram score: as $h$ increases, the dependency between $X_t$ and $f(\Omega_{t-h})$ becomes negligible. Hence, for $h = 3$, we may use the Standard method with the unigram score.

For the General case, we see in the bottom row that the histogram approximately matches the standard normal distribution for $h = 2, 3$ and that the normality assumption always holds. Overall, these results suggest that the assumptions behind the Reprompting method are sound, allowing it (with the $(h+1)$-gram score) to be used for all tested parameter combinations. Hence, all results in §5.2 are computed with the Reprompting method and $(h+1)$-gram score.

### 5.2. Evaluating the spoofing detection tests

To ensure the statistical test is sound, we check whether the Type 1 error rate is properly controlled. This means that, under the null, letting $p$ be the resulting p-value, for all

rejection rates $\alpha \in [0, 1]$,

$$P(p \le \alpha) \le \alpha. \tag{10}$$

We further evaluate the test power on Stealing and Distillation, i.e. how effective it is at distinguishing spoofed text from $\xi$-watermarked text. Additionally, we show in App. H that the Type 1 error rate remains properly controlled in the case of $\xi$-watermarked text that has been edited by humans.

**Type 1 error** To evaluate Type 1 error, we compare the experimental rejection rate under the null hypothesis against the set rate $\alpha$. Per Eq. (10), if the test controls Type 1 error well, we expect the curve to be below the identity function.

In Figure 3, we show the experimental rejection rate of $\xi$-watermarked text on LLAMA2-7B (both instruction fine-tuned and completion models) for different values of $h$ and $T$. We observe that the experimental rejection rates align closely with the identity function. These results show that, in practice, setting a rejection rate of $\alpha$ guarantees that the

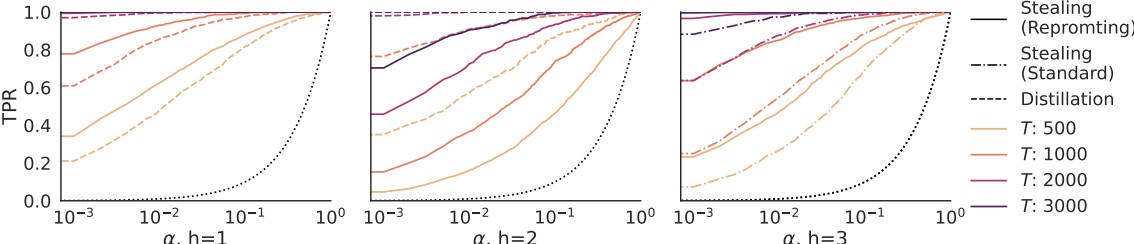

Figure 4: Experimental True Positive Rate of spoofed text. The dotted lines are the identity and serve as a reference for the expected rejection rate under the null. Since, in practice, a low false positive rate ($\alpha$) is desirable, the logarithmic scale on $\alpha$ highlights the true positive rate at low $\alpha$ values.

Table 1: Experimental FPR and TPR for both spoofers at $\alpha \in \{1\%, 5\%\}$, for different $h$ and $T$. $h = 3$ (R) denotes the Reprompting method with $(h + 1)$-gram score while $h = 3$ (S) denotes the Standard method with unigram score. All other entries are for the Reprompting method with $(h + 1)$-gram score.

| Spoofer | | $T = 500$ | | | | $T = 1000$ | | | | $T = 2000$ | | | | $T = 3000$ | | | |
| --- | --- | --- | --- | --- | --- | --- | --- | --- | --- | --- | --- | --- | --- | --- | --- | --- | --- |
| | | FPR @1% | TPR @1% | FPR @5% | TPR @5% | FPR @1% | TPR @1% | FPR @5% | TPR @5% | FPR @1% | TPR @1% | FPR @5% | TPR @5% | FPR @1% | TPR @1% | FPR @5% | TPR @5% |
| STEALING | $h = 1$ | 0.00 | 0.62 | 0.04 | 0.81 | 0.00 | 0.93 | 0.04 | 0.99 | 0.01 | 1.00 | 0.05 | 1.00 | 0.01 | 1.00 | 0.07 | 1.00 |
| | $h = 2$ | 0.01 | 0.16 | 0.04 | 0.35 | 0.00 | 0.37 | 0.04 | 0.59 | 0.01 | 0.73 | 0.05 | 0.88 | 0.01 | 0.91 | 0.04 | 0.97 |
| | $h = 3$ (R) | 0.01 | 0.47 | 0.05 | 0.73 | 0.01 | 0.85 | 0.05 | 0.95 | 0.01 | 0.99 | 0.05 | 1.00 | 0.01 | 1.00 | 0.06 | 1.00 |
| | $h = 3$ (S) | 0.01 | 0.27 | 0.05 | 0.53 | 0.01 | 0.55 | 0.04 | 0.80 | 0.01 | 0.88 | 0.03 | 0.97 | 0.00 | 0.97 | 0.03 | 1.00 |
| DISTILLATION | $h = 1$ | 0.01 | 0.48 | 0.04 | 0.71 | 0.01 | 0.86 | 0.05 | 0.96 | 0.01 | 1.00 | 0.06 | 1.00 | 0.01 | 1.00 | 0.03 | 1.00 |
| | $h = 2$ | 0.01 | 0.57 | 0.06 | 0.78 | 0.01 | 0.91 | 0.06 | 0.97 | 0.01 | 1.00 | 0.05 | 1.00 | 0.00 | 1.00 | 0.07 | 1.00 |

experimental False Positive Rate of the test is indeed $\alpha$.

**Test power**   To evaluate the power of the test, we compute the empirical true rejection rate (i.e., TPR) under the alternative hypothesis for a given threshold $\alpha$.

In Table 1, we provide the experimental False Positive Rate (FPR, rejection under the null) and True Positive Rate (TPR, rejection under the alternative) for a fixed value of $\alpha$. For $T = 3000$, under all tested scenarios, we achieve more than 90% TPR at a rejection rate of 1%. This suggests that, given a long enough text (or concatenation of text), spoofed text from both state-of-the-art methods can be distinguished from $\xi$-watermarked text with high accuracy and reliable control over the false positive rate. Moreover, we see that the Reprompting method yields higher power than the Standard method for all values of $T$. Yet, the Standard method, in the cases where it is applicable, does not require prompting the model $\mathcal{M}$, and thus may still be preferable.

Additionally, in Figure 4, we show the evolution of the TPR with respect to $\alpha$. We observe that for any fixed $\alpha \in [0, 1]$, the power at $\alpha$ converges to 1 as $T$ grows. This indicates that the test can achieve arbitrary TPR at $\alpha$, given sufficiently long text. Also, we see that despite the fundamental differences between the two spoofing techniques, the texts produced by both Stealing and Distillation can be distinguished with the same test. This highlights that the intuition

behind our approach (§3.2) is general and that it points to a fundamental limitation of current spoofing techniques.

## 6. Conclusion

In this work, building on the intuition that spoofed text contains artifacts reflecting the spoofer's knowledge, we successfully constructed statistical tests to distinguish between spoofed and genuine watermarked texts. The tests behave similarly on the studied spoofers, and across a wide range of watermark settings. Our results show that spoofed text can be reliably distinguished from genuine watermarked text with arbitrary accuracy given long enough text, and highlight shared limitations of current learning-based spoofers.

**Limitations**   While we can provide an experimental evaluation of power on current state-of-the-art spoofers, the proposed tests come with no theoretical guarantee of power. We build our tests on reasonable assumptions regarding the limitations of learning-based spoofing techniques. Yet, we hypothesize that spoofing techniques that adaptively learn the vocabulary split may avoid leaving similar artifacts in generated text. Designing such attacks can be an interesting path for future work. Additionally, to have high power, our tests require that the total length of the input texts is not too small. Future work could try to improve the efficiency of our method from this perspective.

# Impact Statement

This paper presents a detailed analysis of texts produced by learning-based spoofing methods and proposes a way to mitigate spoofing issues. While our findings can be used to improve the stealth of future spoofing methods, we believe that improving the understanding of watermark spoofing outweighs the potential misuse of our work.

# Acknowledgments

This work has been done as part of the SERI grant SAFEAI (Certified Safe, Fair and Robust Artificial Intelligence, contract no. MB22.00088). Views and opinions expressed are however those of the authors only and do not necessarily reflect those of the European Union or European Commission. Neither the European Union nor the European Commission can be held responsible for them. The work has received funding from the Swiss State Secretariat for Education, Research and Innovation (SERI) (SERI-funded ERC Consolidator Grant).

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

Table 2: Experimental FPR and TPR for Stealing and Distillation using Dolly instead of C4 as the basis for the generation of $\omega$. The row $h = 3$ (R) corresponds to the Reprompting method with $(h + 1)$-gram score whereas $h = 3$ (S) corresponds to the Standard method with unigram score. Else only the Reprompting method with $(h + 1)$-gram score is used.

| Spoofer | | $T = 200$ | | | | $T = 500$ | | | | $T = 1000$ | | | |
|---|---|---|---|---|---|---|---|---|---|---|---|---|---|
| | | FPR @1% | TPR @1% | FPR @5% | TPR @5% | FPR @1% | TPR @1% | FPR @5% | TPR @5% | FPR @1% | TPR @1% | FPR @5% | TPR @5% |
| STEALING | $h = 1$ | 0.00 | 0.12 | 0.00 | 0.28 | 0.00 | 0.41 | 0.02 | 0.67 | 0.00 | 0.80 | 0.01 | 0.92 |
| | $h = 2$ | 0.00 | 0.03 | 0.04 | 0.15 | 0.00 | 0.12 | 0.04 | 0.28 | 0.01 | 0.34 | 0.08 | 0.50 |
| | $h = 3$ (R) | 0.01 | 0.25 | 0.03 | 0.42 | 0.02 | 0.51 | 0.05 | 0.81 | 0.01 | 0.87 | 0.05 | 0.97 |
| | $h = 3$ (S) | 0.00 | 0.16 | 0.02 | 0.43 | 0.00 | 0.27 | 0.02 | 0.51 | 0.01 | 0.48 | 0.04 | 0.75 |
| DISTILLATION | $h = 1$ | 0.01 | 0.14 | 0.05 | 0.33 | 0.00 | 0.42 | 0.02 | 0.64 | 0.00 | 0.69 | 0.02 | 0.83 |
| | $h = 2$ | 0.02 | 0.12 | 0.07 | 0.27 | 0.01 | 0.36 | 0.07 | 0.59 | 0.02 | 0.67 | 0.07 | 0.84 |

Table 3: Experimental Rejection Rate (RR) for Stealing with SelfHash and $h = 3$ for both $\xi$-watermarked text and spoofed text.

| Experiment | Method | Spoofer LM | $T = 200$ | | $T = 500$ | | $T = 1000$ | | $T = 2000$ | |
|---|---|---|---|---|---|---|---|---|---|---|
| | | | RR @1% | RR @5% | RR @1% | RR @5% | RR @1% | RR @5% | RR @1% | RR @5% |
| $\xi$-watermarked | Reprompting | / | 0.01 | 0.04 | 0.00 | 0.03 | 0.00 | 0.01 | 0.00 | 0.03 |
| | Standard | / | 0.00 | 0.03 | 0.00 | 0.02 | 0.01 | 0.02 | 0.00 | 0.01 |
| STEALING | Reprompting | LLAMA2-7B | 0.12 | 0.30 | 0.31 | 0.59 | 0.70 | 0.90 | 0.99 | 1.00 |
| | | GEMMA-2B | 0.14 | 0.30 | 0.45 | 0.73 | 0.83 | 0.93 | 1.00 | 1.00 |
| | | MISTRAL-7B | 0.10 | 0.29 | 0.38 | 0.63 | 0.79 | 0.93 | 1.00 | 1.00 |
| | Standard | LLAMA2-7B | 0.03 | 0.13 | 0.06 | 0.20 | 0.11 | 0.35 | 0.32 | 0.63 |
| | | GEMMA-2B | 0.03 | 0.14 | 0.07 | 0.26 | 0.15 | 0.40 | 0.36 | 0.62 |
| | | MISTRAL-7B | 0.05 | 0.22 | 0.15 | 0.39 | 0.35 | 0.63 | 0.74 | 0.88 |

# A. Additional Experimental Results

In this section, we conduct several thorough ablation studies. We evaluate the test using a different dataset as base prompts (App. A.1), with a different variation of the watermark scheme (App. A.2), using another watermarked model (App. A.3), using other spoofer models (App. A.4), and using both a different watermarked and spoofer model in App. A.5. In all additional settings tested, the results are similar to those presented in §5, which emphasizes the validity of the test and shows that the spoofing artifacts studied are a fundamental property of learning-based spoofers.

Unlike in §5, we generate 1,000 continuations per parameter combination for the ablation study. It means that on average we have $10^5/T$ samples per parameter combination.

## A.1. Mitigating potential methodological biases

Here, we use the same settings as §5 (Stealing and Distillation with SumHash, different values of $h$, and for $h = 3$, both the Reprompting and Standard methods), but use text continuations of prompts sampled from Dolly (Conover et al., 2023) instead of the C4 dataset. We show that the methodology used to generate the spoofed and $\xi$-watermarked texts has no influence on the results.

In Table 2, we show the experimental FPR and TPR at $\alpha$ of 1% and 5%. The results are similar to those on C4 from Table 1: the Type 1 error is controlled, and the power is similar. This suggests that the methodology we use to generate the prompts does not influence the results. Hence, we can expect that for most texts $\omega$, the empirical results presented hold, and that if $\omega$ is spoofed, the spoofer's artifacts remain present and discoverable.

Table 4: Experimental Rejection Rate (RR) for Stealing with MISTRAL7B as $\mathcal{M}$ at $\alpha$ of 1% and 5% on both $\xi$-watermarked text and spoofed text.

| Experiment | Method | Spoofer LM | $T = 200$ RR @1% | $T = 200$ RR @5% | $T = 500$ RR @1% | $T = 500$ RR @5% | $T = 1000$ RR @1% | $T = 1000$ RR @5% | $T = 2000$ RR @1% | $T = 2000$ RR @5% |
|---|---|---|---|---|---|---|---|---|---|---|
| $\xi$-watermarked | Reprompting | / | 0.02 | 0.05 | 0.03 | 0.08 | 0.00 | 0.04 | 0.00 | 0.02 |
| | Standard | / | 0.01 | 0.04 | 0.01 | 0.02 | 0.00 | 0.02 | 0.00 | 0.02 |
| STEALING | Reprompting | LLAMA2-7B | 0.45 | 0.73 | 0.89 | 1.00 | 0.99 | 1.00 | 1.00 | 1.00 |
| | | GEMMA-2B | 0.48 | 0.90 | 0.97 | 1.00 | 1.00 | 1.00 | 1.00 | 1.00 |
| | | MISTRAL-7B | 0.59 | 0.81 | 0.97 | 1.00 | 1.00 | 1.00 | 1.00 | 1.00 |
| | Standard | LLAMA2-7B | 0.19 | 0.41 | 0.25 | 0.60 | 0.45 | 0.79 | 0.83 | 0.95 |
| | | GEMMA-2B | 0.21 | 0.48 | 0.40 | 0.66 | 0.64 | 0.81 | 0.85 | 0.96 |
| | | MISTRAL-7B | 0.27 | 0.55 | 0.70 | 0.88 | 0.94 | 0.99 | 0.99 | 1.00 |

Table 5: Experimental FPR at $\alpha = 1\%$ and $\alpha = 5\%$ with SumHash $h = 2$, across spoofer LMs. Bold corresponds to the case where both the spoofer and watermarked models are the same.

| Experiment | Spoofer LM | $T = 200$ TPR @1% | $T = 200$ TPR @5% | $T = 500$ TPR @1% | $T = 500$ TPR @5% | $T = 1000$ TPR @1% | $T = 1000$ TPR @5% | $T = 2000$ TPR @1% | $T = 2000$ TPR @5% |
|---|---|---|---|---|---|---|---|---|---|
| STEALING | **LLAMA2-7B** | 0.07 | 0.16 | 0.14 | 0.34 | 0.36 | 0.62 | 0.68 | 0.88 |
| | GEMMA-2B | 0.02 | 0.17 | 0.09 | 0.32 | 0.29 | 0.52 | 0.61 | 0.82 |
| | MISTRAL-7B | 0.05 | 0.16 | 0.16 | 0.35 | 0.37 | 0.59 | 0.73 | 0.88 |
| DISTILLATION | **LLAMA2-7B** | 0.20 | 0.46 | 0.60 | 0.80 | 0.94 | 0.99 | 1.00 | 1.00 |
| | PYTHIA-1.4B | 0.27 | 0.55 | 0.57 | 0.78 | 0.91 | 0.97 | 1.00 | 1.00 |

## A.2. Results for the SelfHash scheme

Next, we focus on SelfHash with $h = 3$ and $\delta = 4$ for Stealing. We use both the Reprompting and the Standard method with their respective score functions (§4.2).

In Table 3, we show the experimental FPR at $\alpha = 1\%$ and $\alpha = 5\%$ for $\xi$-watermarked and spoofed text. Similarly to the SumHash variant, the Type 1 error is properly controlled for both the Standard and the Reprompting methods. Moreover, the empirical power scaling with $T$ is similar to the SumHash scheme from Table 1. This means that the spoofing artifacts are not tied to a specific scheme, but rather represent a fundamental limitation of learning-based watermark spoofing techniques such as Stealing and Distillation. Additionally, we also see that the power of the Standard method at a fixed $T$ is lower than that of the Reprompting method. This confirms the expected trade-off of the unigram score: enforcing cross-independence is traded for power (§4.2).

## A.3. Alternative watermarked model

In this experiment, we use MISTRAL-7B as the watermarked model $\mathcal{M}$ for SumHash at $h = 3$ on Stealing. We do not use a different $\mathcal{M}$ for Distillation, as Distillation was only empirically validated on LLAMA2-7B (Gu et al., 2024).

In Table 4, we show the experimental FPR at $\alpha$ of 1% and 5% for $\xi$-watermarked text and spoofed text on different spoofer LMs. Similar to the results in Table 1, the Type 1 error is controlled in both the Reprompting and Standard methods. Moreover, the power scaling with $T$ is also similar to the results from Table 1. This suggests that the model $\mathcal{M}$ used by the model provider has no influence on the artifacts left by spoofing attempts on such a model.

## A.4. Influence of the spoofer model

In this experiment, we run our tests on SumHash with $h = 2$, using for Stealing LLAMA2-7B, MISTRAL-7B and GEMMA 2B, and for Distillation LLAMA2-7B and PYTHIA-1.4B.

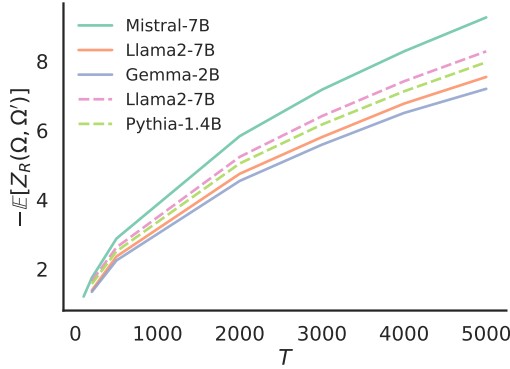

Figure 5: Evolution of $\mathbb{E}[Z_R(\Omega, \Omega')]$ for different spoofer LMs with $T$.

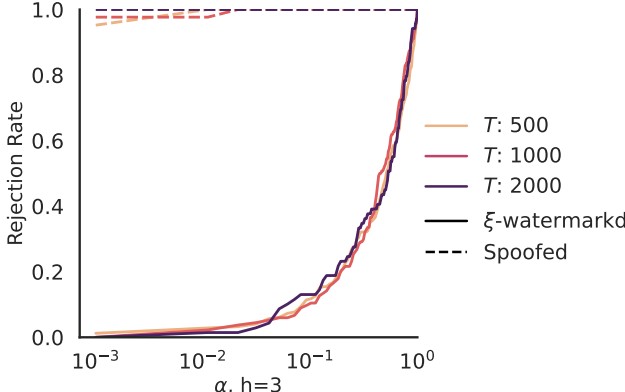

Figure 6: ROC curves with LLAMA3-8B as the watermarked model, *Stealing* with QWEN2.5-7B as the spoofer model, and $h = 3$ with Reprompting.

In Figure 5, we show the evolution of the expected value of $Z_R(\Omega, \Omega')$ for spoofed texts with respect to $T$, across different spoofer LMs. We see that the evolution of the average Z-score is similar across all models and both spoofing techniques. This suggests that the choice of the spoofer LM has almost no influence on the test power.

Additionally, in Table 5, we show the FPR and TPR for the 5 spoofer LMs tested. For $T = 2000$, we obtain similar results across all models, with a TPR at 1% of at least 60% for Stealing and 100% for Distillation, similar to the results from §5.2. Moreover, counterintuitively, a spoofer using the same model as the model owner does not significantly lower the test power. This suggests that the artifacts we are detecting in spoofed text indeed reflect the lack of knowledge of the spoofer (§3.2), and not the difference between the LM used by the spoofer and the LM used by the model provider.

### A.5. Alternative watermarked and spoofer model

In this experiment, we use LLAMA3-8B as the watermarked model $\mathcal{M}$ with SumHash at $h = 3$, and we spoof using Stealing with QWEN2.5-7B as the spoofer model.

Figure 6 shows the ROC curves of our detection test using these alternative models. We see that our test remains valid, with the Type I error being properly controlled (solid line) and the power remaining high (dashed line). This suggests that neither the model $\mathcal{M}$ used by the model provider nor the spoofer model influences the artifacts left by spoofing attempts.

## B. Validating the Concatenation Procedure

In this section, we experimentally validate the claim that concatenating texts $\omega$ according to the procedure from §4.2 has no influence on the resulting distribution of the statistic.

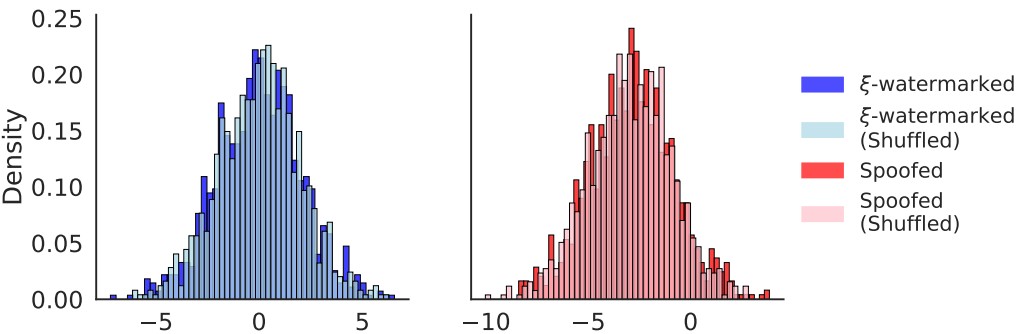

Figure 7: Histogram of Z-scores for both $\xi$-watermarked and spoofed corpora, as well as their shuffled counterparts.

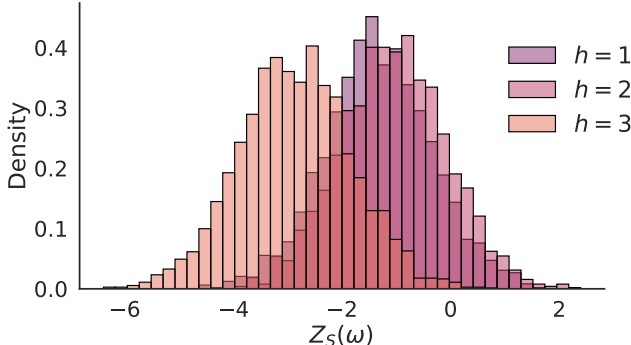

Figure 8: Histogram of $Z_S(W)$ for $\xi$-watermarked text with $(h+1)$-gram score and Standard method.

**Experimental setup**  Let $W = (\omega_1, \ldots, \omega_n)$ be a corpus of $n$ texts of the same length, and $W'$ the corresponding corpus of Reprompting texts of the same length $T$. Let $X, X' \in \{0,1\}^{n \times T}$ be the color matrices of the corpora, and $Y, Y' \in [0,1]^{n \times T}$ be the associated $(h+1)$-gram score matrices of the corpora. For permutations $\sigma \in \mathfrak{S}_{n \times T}$, we define $\sigma(X)_{i,j} = X_{\sigma((i,j))}$ and $\sigma(Y)_{i,j} = Y_{\sigma((i,j))}$. We define $\sigma(W)$ as the *shuffled* corpus with the corresponding $\sigma(X)$ color and $\sigma(Y)$ score. Given $\sigma \in \mathfrak{S}_{n \times T}$, we test the hypothesis that shuffling has no influence on the distribution of $Z_R(W, W')$,

$$Z_R(\sigma(W), \sigma(W')) \sim Z_R(W, W'), \tag{11}$$

where $Z_R(W, W') := (Z_R(\omega_1, \omega'_1), \ldots, Z_R(\omega_n, \omega'_n))$. The shuffling operation can be interpreted as a concatenation of texts of length 1. Hence, if the shuffling has no influence, this implies that the concatenation of texts of longer length has no influence either. To test for the equality of distribution, we use a Mann-Whitney U rank test.

**Results**  In practice, we generate $n = 1000$ $\xi$-watermarked and spoofed texts of length 175 and their corresponding $T = 150$-length Reprompting text corpora. We sample $\sigma \in \mathfrak{S}_{n \times T}$ uniformly in $\mathfrak{S}_{n \times T}$. In Figure 7, we show the resulting histogram of $Z_R(W, W')$ and $Z_R(\sigma(W), \sigma(W'))$. The histograms between the non-shuffled and shuffled versions perfectly overlap for both the $\xi$-watermarked texts and the spoofed texts. Moreover, the resulting p-values from the Mann-Whitney U rank test are 0.86 and 0.44, respectively. Hence, we can conclude that Eq. (11) is verified and that the concatenation procedure has no influence on the distribution of the statistic.

## C. Dependence Between the Context Distribution and the Color

In this section, we study in detail the dependence between the color of token $\omega_t$ and $I_{\mathcal{D}}(\omega_{t-h:t})$ in $\xi$-watermarked text from §3.2.

**Problem statement**  We recall that $\mathcal{D}$ is the training data of the spoofer, and $I_{\mathcal{D}}$ is the function of frequencies of $h+1$-grams in $\mathcal{D}$. In §3.2, we hypothesize that low entropy is a common factor that implies $I_{\mathcal{D}}(\Omega_{t-h:t})$ is high and

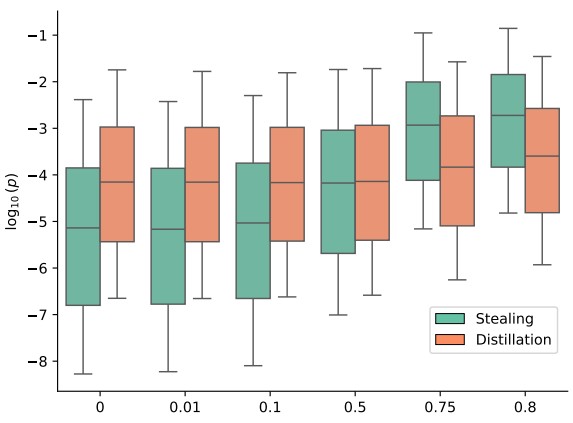
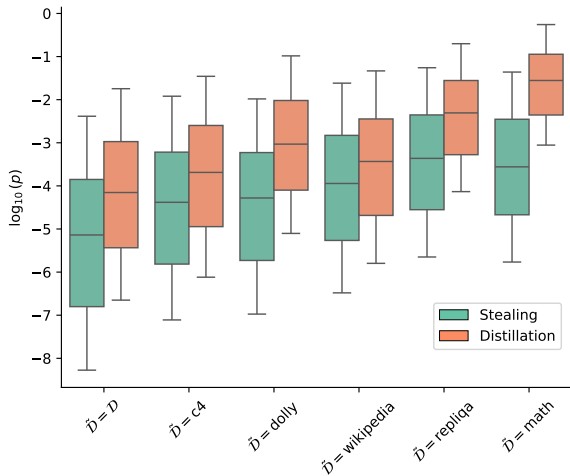

Figure 9: *Left*: Evolution of the p-value distribution with the Total Variation distance between $\mathcal{D}$ and $\tilde{\mathcal{D}}$. Note that the x-scale is not linear. *Right*: Evolution of the p-value distribution for different choices of $\tilde{\mathcal{D}}$. Each p-value is computed with 1000-token long completions. The whiskers are set at $0.5$ of the IQR for visibility.

$P(X_t = 1) \approx \gamma$. Under such an assumption, we therefore expect the correlation between the observed color sequence $x$ and the $(I_{\mathcal{D}}(\omega_{t-h:t}))_{\forall t \in \{h,...,T\}}$ to be negative. In other words, we expect $Z_S(\omega)$ with the $(h+1)$-gram score to be negative for $\xi$-watermarked text.

**Results**    We verify this claim by computing $Z_S(\omega)$ with the $(h+1)$-gram score for a corpus $W$ of 1000 $\xi$-watermarked texts, each of length $T = 500$. In Figure 8, we see the histograms of $Z_S(W)$ for different values of $h$. We see that for all $h$, $\hat{\mathbb{E}}[Z_S(W)]$ is indeed negative. Furthermore, we notice that the histograms appear normally distributed, agreeing with the assumption underlying the Reprompting method (Eq. (5)). Therefore, these results show that the proposed intuitive explanation of the dependence due to $\mathcal{M}$ is coherent, and further highlight the need for the Reprompting method in order to build a statistic with a known distribution when using the $(h+1)$-gram score.

## D. Influence of the Training Dataset

In this section, we study the influence of $\tilde{\mathcal{D}}$ on our ability to detect spoofed texts. As we do not know the true distribution of $\mathcal{D}$, we hope that using a different dataset does not significantly affect our results. We study the influence of $\tilde{\mathcal{D}}$ on both Stealing and Distillation methods with SumHash $h = 1$.

**Evolution with the TV distance**    First, we analyze the influence of the choice of $\tilde{\mathcal{D}}$ in a controlled setting. We let $\tilde{\mathcal{D}}_0$ be the counts of the different $(h+1)$-grams in $\mathcal{D}$. We then build a perturbed dataset $\tilde{\mathcal{D}}_\epsilon$ by adding centered normal noise with standard deviation $\epsilon$ to $\tilde{\mathcal{D}}_0$. Finally, we compute the total variation distance between $\tilde{\mathcal{D}}_\epsilon$ and $\mathcal{D}$. In Figure 9 (left), we observe that the p-values increase on average with the total variation distance between $\tilde{\mathcal{D}}_\epsilon$ and $\mathcal{D}$. This confirms the intuition that the better the estimate of $\mathcal{D}$, the more powerful our tests are. Furthermore, it appears that the p-values increase slowly with the total variation distance, which suggests that the choice of $\tilde{\mathcal{D}}$ is not crucial for obtaining a powerful test.

**Comparing different training datasets**    We run the test for different choices of $\tilde{\mathcal{D}}$ (C4 (Raffel et al., 2020), Dolly (Conover et al., 2023), Wikipedia (WikimediaFoundation), Repliqa (Monteiro et al., 2024), and Math (Fourrier et al., 2023)) as well as $\tilde{\mathcal{D}} := \mathcal{D}$ for comparison. In Figure 9 (right), we see that even the Math dataset has reasonable p-values for Stealing despite our experimental evaluations using significantly different prompt completions from news articles. This confirms that our test is robust to the choice of $\tilde{\mathcal{D}}$.

Lastly, given a received watermarked text $\omega$, a model provider could adjust the choice of $\tilde{\mathcal{D}}$ based on the topic of $\omega$. Such a heuristic could ensure that the choice of $\tilde{\mathcal{D}}$ is always relevant, and further mitigate its impact on the test power.

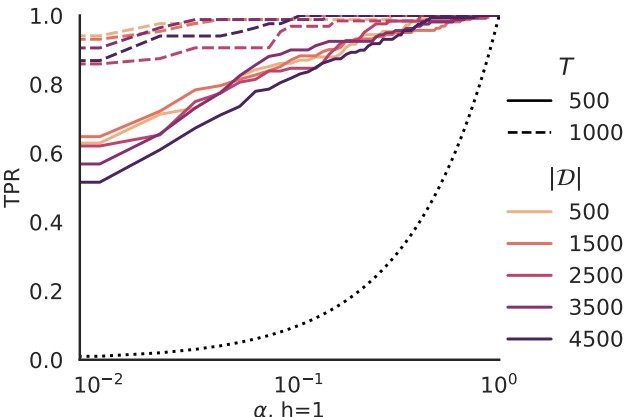

Figure 10: Experimental True Positive Rate of spoofed text with different sizes of $\mathcal{D}$. The size of $\mathcal{D}$ is measured in training steps, where each step comprises $16, 284$ tokens.

## E. Influence of the Size of the Spoofer Training Data

In this section, we study how the power of the test is impacted by the size of the training dataset $\mathcal{D}$. We show that increasing the spoofer training dataset reduces the presence of artifacts at a slow rate, and hence is not an effective way to remove the spoofing artifacts.

We run the test for Distillation with LeftHash, $h = 1$, LLAMA2-7B as the watermarked model, and PYTHIA-1.4B as the spoofer model. We train the spoofer with different sizes of $\mathcal{D}$. We note that, for practicality, the smaller instances of $\mathcal{D}$ are subsets of the larger ones. Otherwise, we run the same experimental procedure as in §5.2, but using only $n = 1000$ completions.

We see in Figure 10 that the influence of increasing the spoofer training dataset $\mathcal{D}$ is very mild. This suggests that, even though increasing the spoofer training data indeed lowers the power of the test, the rate at which it does so is so slow that it is not an effective way to hide spoofing artifacts. Indeed, a 9-time increase in the size of $\mathcal{D}$ only reduced the TPR at 0.1 percent from 94 percent to 87 percent with $T = 1000$.

## F. Influence of the Reprompting

In this section, we study in greater detail the impact of the Reprompting method. Specifically, in App. F.1 we find that estimating the prompts compared to using the original prompts only has a minor impact, and in App. F.2 we show that using a fixed corpus to estimate the correlation directly (i.e., $\mu_\Omega$ from Eq. (5)) leads to a non-controllable Type 1 error. Hence, the Reprompting method is needed to, in practice, properly control the Type 1 error.

### F.1. Impact of Estimating the Prompts

In this part, we analyze whether the distribution shift between the original prompts and the estimated prompts significantly influences our test power.

We run the detection test using the Reprompting statistic (Eq. (6)) once with the original prompt and once without. Otherwise, we use LLAMA2-7B as the watermarked model and MISTRAL-7B as the spoofer model with Stealing, $h = 1$, and the same experimental setup as in §5. Figure 11 (left) shows the ROC curve with and without using the original prompts. As expected, we find that estimating the prompt consistently slightly degrades the TPR (at worst a 5% TPR at 1% FPR decrease). This suggests that, for non-adversarial cases, the drift incurred by using the beginning of the received text as a proxy for the prompt has minimal impact on our detection accuracy.

### F.2. Using a Fixed Corpus Instead of Reprompting

In this part, we show that using a fixed estimate of the correlation distribution for any text leads to an uncontrollable Type I error, hence justifying why the Reprompting method is needed.

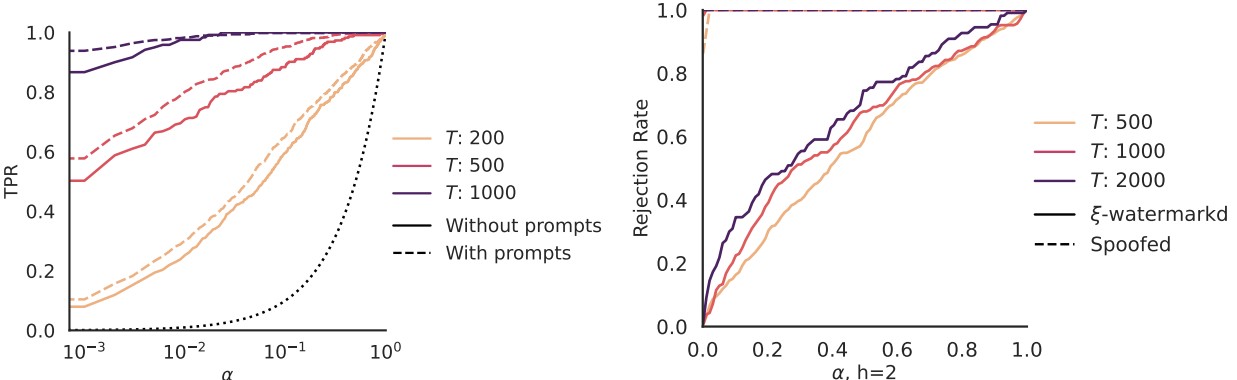

Figure 11: (Left) Influence of the prompt estimation with Reprompting: ROC curves with LLAMA2-7B as the watermarked model, *Stealing* with MISTRAL-7B as the spoofer model, and $h = 1$. The dotted black line corresponds to the identity function as a reference. (Right) ROC curves with LLAMA2-7B as the watermarked model, *distillation* with LLAMA2-7B as the spoofer model, and $h = 2$, using a fixed corpus of $\xi$-watermarked text to estimate the average watermarked correlation.

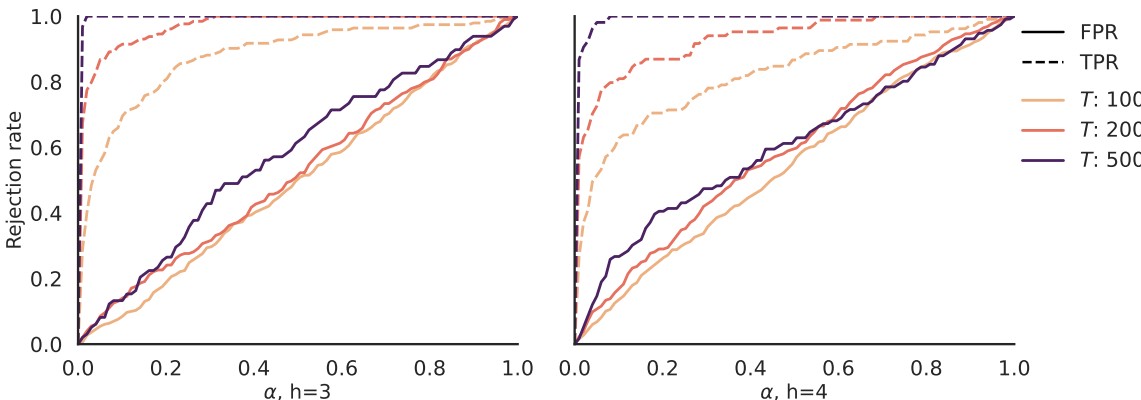

Figure 12: Rejection rates for the Reprompting method on the AAR watermark with $h = 3$ and $h = 4$. The solid lines correspond to $\xi$-watermarked text and the dashed lines to Distillation-spoofed text.

We first generate a corpus of 512k tokens of $\xi$-watermarked text using LLAMA2-7B as the watermarked model and $h = 2$, generated using 50-token-long prompts from OPENWEBTEXT. For a text $\omega$, to compute a p-value, we directly apply Eq. (6), where we replace the "reprompted correlation" $S(\omega')$ with the fixed correlation estimate from the corpus, independent of $\omega$. In Figure 11 (right), we show the ROC curve of such modified detection text using Distillation with LLAMA2-7B as a spoofer. We see that the FPR is higher than what we would expect, suggesting that the estimated mean does not capture the true mean. This is why Reprompting, albeit more costly, is more reliable in practice.

## G. Extending the Method to Other Schemes

While we design our method to detect spoofing attempts on Red-Green schemes (Kirchenbauer et al., 2023), as these are the primary target of several spoofing works, we show that the method can be generalized to other watermarking schemes. Excluding the unigram scheme by Zhao et al. (2024), which Zhang et al. (2024) shows can be perfectly spoofed, we can study the AAR scheme from Aaronson (2023), as well as one of the KTH schemes from Kuditipudi et al. (2024), as both schemes were shown to be spoofable via Distillation (Gu et al., 2024).

**AAR watermark** In the AAR watermark, $h$ previous tokens are hashed using a private key $\xi$ to obtain a score $r_i$ uniformly distributed in $[0, 1]$ for each token index $i$ in the vocabulary $\Sigma$. Given $p_i$, the original model probability for token index $i$, the next token is then deterministically chosen as the token $i^*$ that maximizes $r_i^{1/p_i}$.

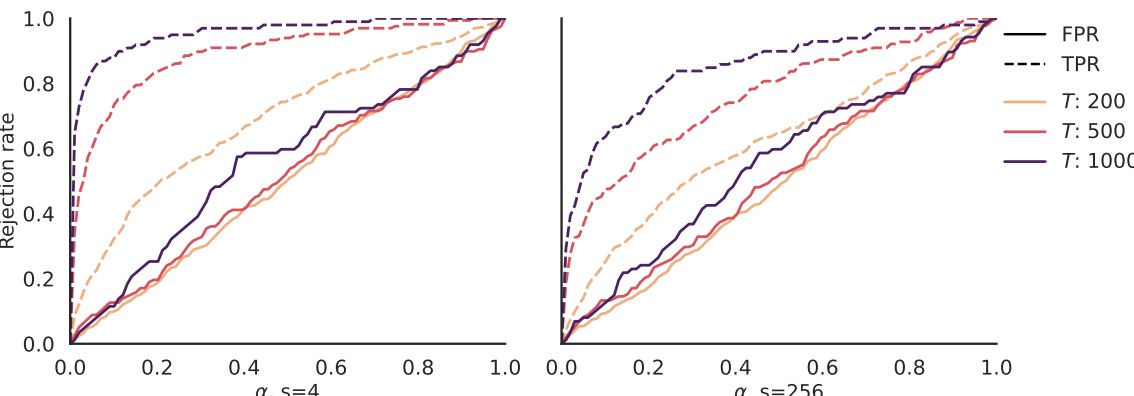

Figure 13: Rejection rates for the Reprompting method on the KTH watermark with $s \in \{1, 4, 256\}$. The solid lines correspond to $\xi$-watermarked text and the dashed lines to Distillation-spoofed text.

Given a text $\omega \in \Sigma^T$, we naturally generalize Eq. (4) by defining $x \in \mathbb{R}^T$ as $x_t = -\log r_{\omega_t}$, whereas previously $x_t$ was the color of the $t$-th token. The rest of the method remains identical.

We evaluate both the FPR and TPR of our test using $h \in \{3, 4\}$, LLAMA2-7B as both the watermarked model and the attacker model, the Reprompting method, and the same experimental procedure as in §5, except that we generate only $n = 500$ completions. We discarded $h = 2$ as the watermarked model output was too low-quality and repetitive (Gu et al., 2024). In Figure 12, we see that the generalized method can successfully detect spoofed text with a 90% TPR at a rejection rate of 1% for 500 tokens. In fact, it is even more powerful than the detection in the Red-Green scheme, where we achieved a similar TPR at 1% with 3000 tokens (§5.2). However, the test hypothesis appears slightly violated, as the empirical FPR at 1% is around 2% for both $h = 3$ and $h = 4$.

**KTH watermark** In the KTH watermark (EXP variant), a single watermark key sequence of length $n_{key}$, $\xi = \xi^1, \ldots, \xi^{n_{key}}$, is uniformly distributed, where each $\xi^i \in [0, 1]^{|\Sigma|}$. To generate the $j$-th token (modulo $n_{key}$), the watermark samples the token $i^*$ that maximizes $\left(\xi_i^j\right)^{1/p_i}$. Additionally, to allow more diversity in the generated text, the key is randomly shifted by a constant at each query. As in Gu et al. (2024), we denote by $s$ the number of allowed shifts.

Given a text $\omega \in \Sigma^T$, we naturally generalize Eq. (4) by defining $x \in \mathbb{R}^T$ as $x_t = \log(1 - \xi_{\omega_t}^t)$, whereas previously $x_t$ was the color of token $t$. To account for the permutation of the key, we further replace $\log(1 - \xi_{\omega_t}^t)$ with the Levenshtein cost introduced in Kuditipudi et al. (2024). Moreover, the scheme, being based on a fixed key, lacks any context $h$ that can be used to compute the N-gram score $y_t$ (Eq. (7)). Following the intuition from Gu et al. (2024) that, in the limit, their spoofing ability comes from learning contiguous watermarked sequences of length $n_{key}$, we suspect that setting $h \approx n_{key}$ would enable greater test power. In practice, due to practical constraints, we set $h = 5$. The rest of the method remains unchanged.

We evaluate both the FPR and TPR of our test, using $s = 4$, and $s = 256$, along with a key of length $n_{key} = 256$, on LLAMA2-7B as both the watermarked model and the attacker model, the Reprompting method, and the same experimental procedure as in §5, except that we generate only $n = 500$ completions. In Figure 13, we see that this generalized method can successfully detect spoofed text for both $s = 4$ and $s = 256$, albeit with a TPR of 65% at a confidence of 99% for $s = 4$ and TPR of 30% for $s = 256$. In all three cases however, the Type 1 error is controlled, i.e., empirical FPR corresponds to the theoretical FPR.

## H. Influence of Human Modifications on FPR

Here, we study the behavior of $\xi$-watermarked text that has subsequently been edited by humans. We consider two different use cases. The first is the case of cropping. Given a $\xi$-watermarked text, we assume a human inserts non-watermarked text in the middle. This corresponds to a plausible use case of LLMs, where humans merge generated text with their own. Second, we consider paraphrasing. Given a $\xi$-watermarked text, we paraphrase it using DIPPER (Krishna et al., 2023).

We evaluate the FPR of human-modified text using $h = 3$ on both the Standard and Reprompting methods and follow a

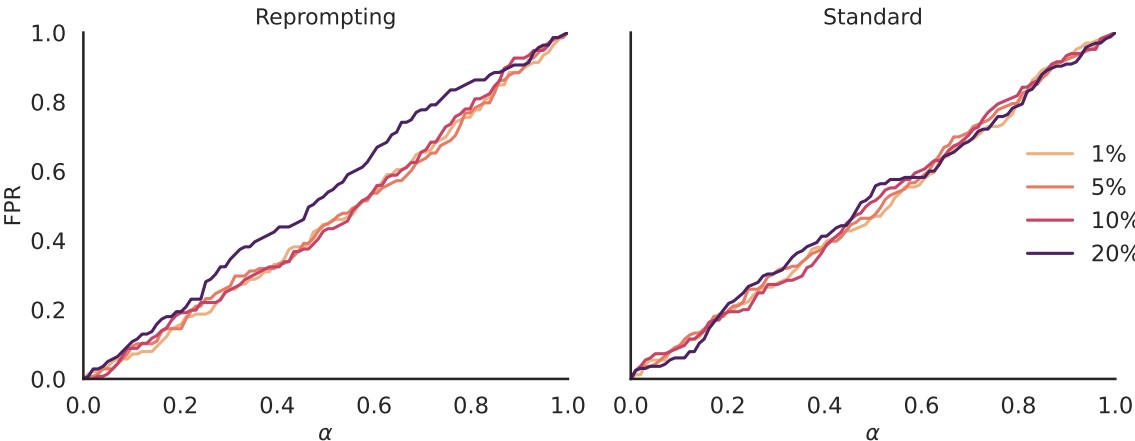

Figure 14: Experimental rejection rate of mixed $\xi$-watermarked text and human text on LLAMA2-7B for both the Reprompting method (left) and the Standard method (right) at different percentages of human text. Each mixed text is in total $500$ tokens long.

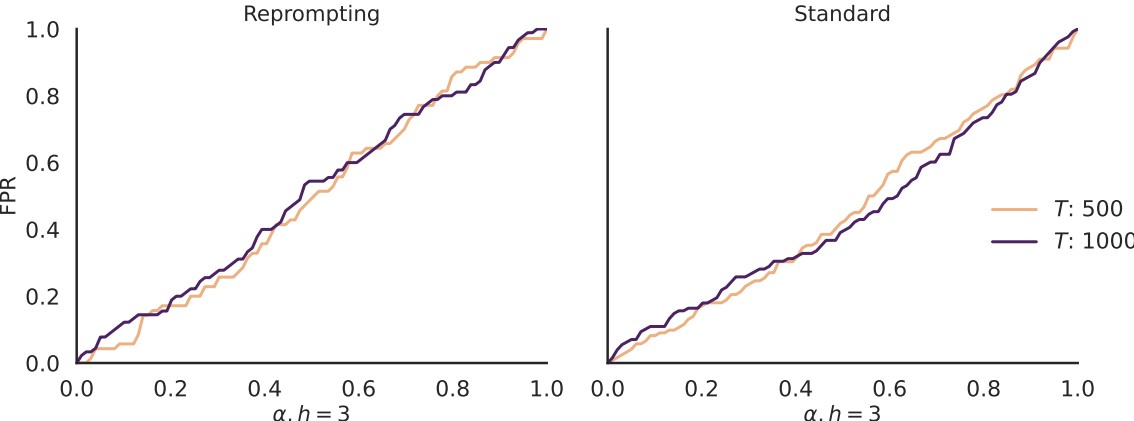

Figure 15: Experimental rejection rate of paraphrased $\xi$-watermarked text on LLAMA2-7B for both the Reprompting method (left) and the Standard method (right).

similar experimental procedure as in §5, except that we generate only $n = 500$ samples. Given a percentage $\rho$, for each generated C4 prompt completion of length $T$, we randomly insert another random human text sampled from C4 such that $\rho$ percent of the resulting text is human-generated. We used this procedure for $\rho \in \{0.01, 0.05, 0.1, 0.2\}$. As in §5, we apply the test only on text that appears watermarked according to the original watermark detector. In Figure 14, we see that even for the highest percentage of human text (20%), the test properly controls Type 1 error.

We evaluate the FPR of the paraphrased text using $h = 3$ on both the Standard and Reprompting methods and follow a similar experimental procedure as in §5, except that we generate only $n = 1000$ samples. We note that we apply the test only on text that is considered watermarked by the original watermark detector. In Figure 15, we see that the test still properly controls Type I error for both methods and for different text lengths.

Both results show that a rejection rate of $\alpha$ still guarantees an experimental FPR of $\alpha$, even if the $\xi$-watermarked texts have been altered by humans.

# I. Proof of Lemma 4.1

In this section, we detail the proof of Lemma 4.1.

First, let's recall some statistical results that we need.

**Theorem I.1** (Lindeberg CLT). *Let $X_{n,1}, ..., X_{n,n}$ be independent random variables in $\mathbb{R}^d$ with mean zero. If for all $\varepsilon > 0$*

$$\sum_{k=1}^{n} \mathbb{E}[||X_{n,k}||^2 \mathbb{1}\{||X_{n,k}|| > \varepsilon\}] \to 0, \text{ (Lindeberg Condition)} \tag{12}$$

*and*

$$\sum_{k=1}^{n} cov(X_{n,k}) \to V, \tag{13}$$

*then*

$$\sum_{k=1}^{n} X_{n,k} \xrightarrow{d} \mathcal{N}(0, V). \tag{14}$$

**Theorem I.2** (Delta method). *Let $X_1, ..., X_n$ be a sequence of random variables in $\mathbb{R}^d$, if*

$$\sqrt{n}(X_n - \mu) \xrightarrow{d} \mathcal{N}(0, V), \tag{15}$$

*and $u : \mathbb{R}^d \to \mathbb{R}$ is differentiable at $\mu$, with $\nabla u(\mu) \neq 0$, then*

$$\sqrt{n}(u(X_n) - u(\mu)) \xrightarrow{d} \mathcal{N}(0, \nabla u(\mu)^T V \nabla u(\mu)). \tag{16}$$

Now we proceed to prove Lemma 4.1. We first state the result formally.

**Lemma 4.1.** *Let $X := X_1, \ldots, X_T$ be a sequence of independent (non i.i.d) Bernoulli random variables, and $g_i = P(X_i = 1)$. Let $Y := Y_1, \ldots, Y_T$ be a sequence of i.i.d. random variables. Let $\Omega = (X, Y)$. Assuming that, for all $i \in \{0, \ldots, T\}$, $X_i$ and $Y_i$ are independent, that there exist $g^{(1)}, g^{(2)} \in [0, 1]$ such that*

$$\lim_{T \to \infty} \frac{1}{T} \sum_{i=1}^{T} (g_i - g^{(1)}) = O\left(\frac{1}{T}\right) \text{ and } \lim_{T \to \infty} \frac{1}{T} \sum_{i=1}^{T} g_i^2 = g^{(2)}, \tag{17}$$

*and assuming that $Y$ admits at least 4 moments $\mu_Y, \mu_{Y^2}, \mu_{Y^3}, \mu_{Y^4}$. Then, we have that*

$$Z_S(\Omega) := \sqrt{T} S(\Omega) \xrightarrow{d} \mathcal{N}(0, 1).$$

*Proof.* Let $w_i := (X_i, Y_i, X_i^2, Y_i^2, X_i Y_i)$. Let $X_{n,k} = \frac{(w_i - \mathbb{E}[w_i])}{\sqrt{n}}$. We recall the definition of $S$,

$$S(\Omega) = \frac{\sum_{t=1}^{T} (X_t - \bar{X}_T)(Y_t - \bar{Y}_T)}{\sqrt{\sum_{t=1}^{T} (X_t - \bar{X}_T)^2 \sum_{t=1}^{T} (Y_t - \bar{Y}_T)^2}}. \tag{18}$$

$$= \frac{\frac{1}{T} \sum_{t=1}^{T} X_t Y_t - \left(\frac{1}{T} \sum_{t=1}^{T} X_t\right)\left(\frac{1}{T} \sum_{t=1}^{T} Y_t\right)}{\sqrt{\frac{1}{T} \sum_{t=1}^{T} X_t^2 - \left(\frac{1}{T} \sum_{t=1}^{T} X_t\right)^2} \sqrt{\frac{1}{T} \sum_{t=1}^{T} Y_t^2 - \left(\frac{1}{T} \sum_{t=1}^{T} Y_t\right)^2}}, \tag{19}$$

where $\bar{X}_T$ denotes the mean of $X_{1:T}$.

The proof goes as follows:

- First, we show that the sum of the covariance matrix of $X_{n,k}$ converges (Eq. (13)).

- Then, we show that $X_{n,k}$ satisfies the Lindeberg condition (Eq. (12)). We can then apply the Lindeberg theorem to show that $w_i$ converges to a normal distribution.

- Finally, we apply the Delta method (Theorem I.2) to show that $S(\Omega)$ is normally distributed.

We have that for all $i \neq j$, $w_i$ is independent of $w_j$. For each $i$, we have

$$\text{Cov}(w_i) = \begin{bmatrix} g_i(1-g_i) & 0 & g_i(1-g_i) & 0 & \mu_Y g_i(1-g_i) \\ 0 & -\mu_Y^2+\mu_{Y^2} & 0 & -\mu_Y\mu_{Y^2}+\mu_{Y^3} & g_i(-\mu_Y^2+\mu_{Y^2}) \\ g_i(1-g_i) & 0 & g_i(1-g_i) & 0 & \mu_Y g_i(1-g_i) \\ 0 & -\mu_Y\mu_{Y^2}+\mu_{Y^3} & 0 & -(\mu_{Y^2})^2+\mu_{Y^4} & g_i(-\mu_Y\mu_{Y^2}+\mu_{Y^3}) \\ \mu_Y g_i(1-g_i) & g_i(-\mu_Y^2+\mu_{Y^2}) & \mu_Y g_i(1-g_i) & g_i(-\mu_Y\mu_{Y^2}+\mu_{Y^3}) & g_i(-\mu_Y^2 g_i+\mu_{Y^2}) \end{bmatrix},$$

where we denote $\mu_{Y^k}$ as the $k$-th moment of $Y$.

Then, using Eq. (17), we have that $1/T \sum_{i=1}^{T} \text{Cov}(w_i) = \sum_{i=1}^{T} \text{Cov}(X_{n,i})$ converges towards $V \in \mathbb{R}^{5\times 5}$, defined as

$$V = \begin{bmatrix} g^{(1)}-g^{(2)} & 0 & g^{(1)}-g^{(2)} & 0 & \mu_Y(g^{(1)}-g^{(2)}) \\ 0 & -\mu_Y^2+\mu_{Y^2} & 0 & -\mu_Y\mu_{Y^2}+\mu_{Y^3} & g^{(1)}(-\mu_Y^2+\mu_{Y^2}) \\ g^{(1)}-g^{(2)} & 0 & g^{(1)}-g^{(2)} & 0 & \mu_Y(g^{(1)}-g^{(2)}) \\ 0 & -\mu_Y\mu_{Y^2}+\mu_{Y^3} & 0 & -(\mu_{Y^2})^2+\mu_{Y^4} & g^{(1)}(-\mu_Y\mu_{Y^2}+\mu_{Y^3}) \\ \mu_Y(g^{(1)}-g^{(2)}) & g^{(1)}(-\mu_Y^2+\mu_{Y^2}) & \mu_Y(g^{(1)}-g^{(2)}) & g^{(1)}(-\mu_Y\mu_{Y^2}+\mu_{Y^3}) & -\mu_Y^2 g^{(2)}+\mu_{Y^2}g^{(1)} \end{bmatrix}.$$

We have completed the first step of the proof.

Now we want to show that $X_{n,i}$ satisfies the Lindeberg condition (Eq. (12)). Let $\varepsilon > 0$. Because $X_i, Y_i \in [0,1]$, we have that for all $i \leq n$, $\|X_{n,i}\| \leq \sqrt{\frac{10}{n}}$. There exists $n_0 > 0$ such that $\forall n \geq n_0$, $\sqrt{\frac{10}{n}} < \varepsilon$. Therefore, $\forall n \geq n_0, \forall k \leq n$, $\mathbb{1}\{\|X_{n,k}\| > \varepsilon\} = 0$. So, for all $n \geq n_0$,

$$\sum_{k=1}^{n} \mathbb{E}[\|X_{n,k}\|^2 \mathbb{1}\{\|X_{n,k}\| > \varepsilon\}] = 0. \tag{20}$$

Hence, we have shown that for all $\varepsilon > 0$,

$$\sum_{k=1}^{n} \mathbb{E}[\|X_{n,k}\|^2 \mathbb{1}\{\|X_{n,k}\| > \varepsilon\}] \to 0. \tag{21}$$

Therefore, using the Lindeberg CLT (Theorem I.1), we have that

$$\frac{1}{\sqrt{T}} \sum_{i=1}^{T} (w_i - \mathbb{E}(w_i)) \xrightarrow{d} \mathcal{N}(0, V). \tag{22}$$

We have completed the second step of the proof. Now, we want to apply the Delta method (Theorem I.2) to show that $S(\omega)$ is normally distributed.

Let $\mu_w := \lim_{T \to \infty} 1/T \sum_{i=1}^{T} \mathbb{E}[w_i] = (g, \mu_Y, g, \mu_{Y^2}, g\mu_Y)$. We introduce

$$E_i = \frac{1}{\sqrt{T}} \sum_{i=1}^{T} \mathbb{E}[w_i] - \mu_w \tag{23}$$

$$= \sqrt{T} \left( \frac{1}{T} \sum_{i=1}^{T} \mathbb{E}[w_i] - \mu_w \right) \tag{24}$$

$$= O\left(\frac{1}{\sqrt{T}}\right) \text{ (Using Eq. (17)).} \tag{25}$$

Therefore, we have

$$\frac{1}{\sqrt{T}} \sum_{i=1}^{T} (w_i - \mu_w) = \frac{1}{\sqrt{T}} \sum_{i=1}^{T} (w_i - \mathbb{E}[w_i]) + E_i \xrightarrow{d} \mathcal{N}(0, V). \tag{26}$$

Let $u : \mathbb{R}^5 \to \mathbb{R}$ be defined as

$$u(x) = \frac{x_5 - x_1 x_2}{\sqrt{(x_3 - x_1^2)(x_4 - x_2^2)}}. \tag{27}$$

We have that $S(\Omega) = u\left(1/T \sum_{i=1}^{T} w_i\right)$ (using Eq. (19)) and $u(\mu_w) = 0$, and therefore using the Delta method (Theorem I.2) we have that

$$\sqrt{T}S(\Omega) \xrightarrow{d} \mathcal{N}\left(0, \nabla u(\mu_w)^T V \nabla u(\mu_w)\right). \tag{28}$$

Because $\nabla u(\mu_w)^T V \nabla u(\mu_w) = 1$, we have shown that

$$\sqrt{T}S(\Omega) \xrightarrow{d} \mathcal{N}(0, 1). \tag{29}$$

$\square$

## J. Extended Discussion of the State of Watermark Spoofing

In this section, we overview of the state of the field of watermark spoofing to further motivate our work and highlight its practical implications. In App. J.1, we identify three categories of spoofing techniques and highlight learning-based spoofing techniques as the most practically relevant. We put our findings in this context, discussing the potential for adaptive spoofing that does not leave artifacts in the spoofed text. In App. J.2 we discuss how latest schemes attempt to tackle the issue of spoofing.

### J.1. Approaches to spoofing

**Learning-based spoofing**    As explained in §1, *learning-based spoofing* operates in two phases. In the first phase, the spoofer queries the model to generate a dataset $\mathcal{D}$ of $\xi$-watermarked text. From this dataset $\mathcal{D}$, the spoofer learns the watermark, which allows them to generate spoofed text. In the second phase, using their knowledge and a private LM, the spoofer can generate *arbitrary* watermarked text *at scale*, without having to query the original model again. In particular, spoofed texts can be created as answer to any prompt, even the one that would be refused by the original LLM, which gives learning-based spoofers great flexibility, and illustrates the potential threat they pose. Additionally, as long as the cost of the first phase is reasonable, learning-based spoofing is cost-effective, as the subsequent per-spoofed-text cost is zero. Learning-based spoofing includes the works of Jovanović et al. (2024); Gu et al. (2024); Zhang et al. (2024).

**Piggyback spoofing**    As discussed in §2, the second family of spoofing techniques is *piggyback spoofing*, introduced by (Pang et al., 2024), which directly exploits the desirable robustness property of the watermarks. Given a $\xi$-watermarked sentence, the attacker modifies a few tokens to alter the meaning of the original sentence while maintaining the watermark, interpreting the result as an instance of spoofing. While illustrating the potential drawbacks of high robustness, this comes with several caveats. First, abusing the robustness of the watermark naturally raises the question of the boundary between spoofed text and edited $\xi$-watermarked text. Indeed, mixing human and LM text is a realistic use of LMs, and it is agreed that watermarks should account for this use (Kirchenbauer et al., 2023; Kuditipudi et al., 2024). Second, piggyback spoofing is limited in the scope of text it can generate, as it relies on the original model to generate the majority of the text. This greatly reduces the flexibility of the attack, i.e., does not allow the attacker to generate texts on harmful topics that would be refused by the watermarked model. Finally, the same property makes the cost of spoofing scale with the number of spoofed texts, as the attacker needs to query the original model each time.

**Step-by-step spoofing**    Finally, as also briefly mentioned in §2, a third category of spoofing techniques is *step-by-step spoofing*. This line of works considers spoofing techniques that require queries *at each step* of the generation process of *every spoofed text* (Pang et al., 2024; Zhou et al., 2024; Wu & Chandrasekaran, 2024), using the feedback obtained this way to choose the next token. While they have higher flexibility compared to piggyback spoofing, a key limitation of these techniques is the high cost, even compared to piggyback spoofing. Further, some of these methods assume access to the watermark detector itself (sometimes also its confidence score) to obtain the desired feedback, which is not always realistic. For instance, in the case of the first public large-scale deployment of a watermark, SynthID-Text, Google does not provide public access to the watermark detector.

**Summary**    In summary, learning-based spoofing is the most practically relevant category of spoofing techniques, as it is cost-effective, flexible, and does not require querying the original model for each spoofed text. Another advantage from the perspective of our research question is the fact that current learning-based methods are based on fundamentally different principles, making the question of their common limitations relevant and interesting. In this work, we study that question,

showing that all learning-based spoofers leave visible artifacts in spoofed text, which can be leveraged to distinguish between spoofed and $\xi$-watermarked text.

## J.2. Spoofing-aware watermarking schemes

The field of watermarking is evolving rapidly, as explained in §2, with different schemes proposed in the literature. We distinguish two approaches to watermarks in LMs. The first one is the statistical approach, notably including schemes from Kirchenbauer et al. (2023); Kuditipudi et al. (2024); Aaronson (2023), which place great emphasis on watermark robustness and practicality. The second is the cryptographic approach, with schemes stemming from Christ et al. (2024b), which focus primarily on watermark security and rigorous guarantees.

In particular, schemes with cryptographic features have not been shown to be vulnerable to spoofing attacks. Yet, they enhance security by trading off other key watermark properties, such as robustness to watermark removal. Moreover, recent work (Zhou et al., 2024) suggests merging both fields to create a watermarking scheme that is not only more robust to watermark removal but also to watermark spoofing. However, they show that their approach trades off with generation quality. This highlights that, from the perspective of a model provider, there is no single scheme that is the most desirable. Hence, choosing a particular scheme is a complex task that involves navigating the tradeoffs between different properties. From this perspective, our work provides new evidence that schemes derived from (Kirchenbauer et al., 2023) are harder to spoof than previously thought (as these attempts can be detected by observing the artifacts), and can help model providers adjust their expectations.

