# OpenReview forum: "Discovering Spoofing Attempts on Language Model Watermarks"
_ICML.cc/2025/Conference — ICML 2025 poster_

### Official Review · Reviewer_p27M · 2025-03-06

**Overall Recommendation:** 3

**Summary:**

This paper proposes a statistical test for identifying spoofing attacks against sampling-based LLM watermarks. The method is based on the intuition that the frequency of watermark violation conditioned different n-grams in spoofed texts would be different with that in actual watermarked texts. The result shows that the statistic can indeed have a good performance in spoofed text detection.

**Claims And Evidence:**

The key contribution claimed by the paper includes the in-depth analysis of artifacts in spoofed texts, statistical tests to distinguish spoofed texts and empirical evaluation on the proposed methods. These claims are supported by the theoretical analysis and empirical evaluations in the paper.

**Essential References Not Discussed:**

As far as I know, the references are adequately discussed.

**Experimental Designs Or Analyses:**

The model evaluates the spoofed text detection performance with two different techniques (stealing and distillation) on different models. The experiment designs and analyses look good to me. One minor concern is that the used models are rather older models (Llama2, Gemma) and there have been more advanced models in recent six months (Llama3, Gemma2).

**Methods And Evaluation Criteria:**

The key intuition behind the method is that the spoofing attack will only learn the vocabulary split based on existing corpus, so that the frequency of watermark violation in the spoofed text, conditioned on the different n-grams, will be different with that in the actual watermarked text. This idea is novel and makes sense for the spoofing attack with stealing and distillation (in the case of distillation the learning is also based on the frequency of different n-grams). Nevertheless, it is worth noticing that the it is limited to certain spoofing attacks and may not work against, for example, generating watermarked text by the model and modifying it manually to get spoofed texts.

**Other Comments Or Suggestions:**

N/A

**Other Strengths And Weaknesses:**

The paper does not discuss the possibility of adversarial attacks, where the adversary knows the idea of this statistic and aim to bypass it. I think that it would not be difficult for a knowledged adversary to bypass the statistic, for example, by enforcing the n-grams appearing in the spoofing learning process to be uniform.

**Questions For Authors:**

Do you have any mitigation if the adversary knows your statistics and intentionally bypasses it?

**Relation To Broader Scientific Literature:**

This paper is among the first to study the spoofing attacks in LLM watermarks and propose a statistic for detection. This helps with the robustness of LLM watermark to be applied in practice.

**Theoretical Claims:**

I did not check the mathematical details of the proofs, but the they make intuitive sense to me.

---

> ### Author Rebuttal · Authors · 2025-03-31
>
> We thank the reviewer for their helpful feedback and address their individual questions below. We have attached one additional figure [here](https://drive.google.com/file/d/1Rh3JLVob1UuV2-ElQ7_QgnDNi2CZ-mvb/view).
>
> **Q1: Can the current spoofing methods be *easily* adapted to bypass the proposed detection? For instance, can we constrain the h-gram distribution in the spoofer’s training data?**
>
> We do not believe so, as we are tackling a fundamental limitation of learning-based spoofers that is not specific to a given implementation. Because the spoofer learns from a finite, small training dataset (for cost reasons), it cannot “see” every h-gram combination and hence perfectly reproduce the watermark. Enforcing uniformity is not possible, as there are roughly O($\Sigma^{h-1}$) h-gram combinations (for a fixed last token), which means that a uniform dataset needs an exponential amount of tokens.
>
> In particular, for Stealing, their authors explain that the sparsity of the training data was the main challenge behind their attack, and they had to discard low-frequency h-grams for stability [1]. Enforcing additional constraints on the training data, such as trying to get a distribution of h-grams as uniform as possible, would only reinforce the sparsity issue, ultimately hurting spoofing performance.
>
> For Distillation, we show in App. E Figure 9 that increasing the spoofer's training data by 10x only reduces our method's TPR by 15\%. Further, enforcing constraints on the distribution of the data in that case might significantly degrade the spoofer model's performance. Because the token distribution would deviate further from human text, it might hinder the finetuning process.
>
> Overall, while we cannot provide guarantees against (arbitrary) adversarial attacks, our method practically increases the amount of effort needed by an adversary for successful spoofing—which we argue is overall  beneficial.
>
> **Q2: Does there exist an adversary that could bypass the proposed method?**
>
> As briefly discussed in Section 2 and Appendix I, we believe that a step-by-step spoofing attack, combined with the idea of color-aware substitution [2], would result in a spoofing attack that is not detectable by our method. We note that such an adversary would (in the limit) be able to accurately estimate the Red-Green splits for any context, resulting in a watermark signal that is distributionally indistinguishable from the genuine watermarking algorithm.
>
> However, such a method would require querying the watermarked model for *every* token generated by the spoofer, rendering it prohibitively expensive and impractical. Hence, we argue that, given the current state of the field, there exists no spoofing method that could leverage the knowledge of our statistics to bypass our method while maintaining similar properties (cost and practicality) to current learning-based spoofers.
>
> **Q3: Why do the authors restrict themselves to older models?**
>
> We evaluate our method on older models to match the experimental setup from the spoofing attacks papers. Nonetheless, in response to the reviewers comments, we evaluate Stealing with $h=3$ using *Llama3-8B* as the watermarked model and *Qwen2.5-7B* as the spoofer model, using the Reprompting method (see attached Figure 1). We see that our test is equally valid with newer models as well, with the FPR being properly controlled (solid line) and the TPR high (dashed line).
>
> [1] https://arxiv.org/abs/2402.19361 \
> [2] https://arxiv.org/abs/2403.14719

---

### Official Review · Reviewer_nCP1 · 2025-03-08

**Overall Recommendation:** 3

**Summary:**

One use of watermarking schemes for generative models is for attribution. In recent years there have been several "spoofing" attacks on watermarking schemes which can make a certain piece of text to appear as if it was generated by a certain model by "copying" the watermark associated with that model. This paper comes up with an empirical technique to detect such spoofing attacks.

**Claims And Evidence:**

Yes

**Essential References Not Discussed:**

See "strengths/weaknesses" below.

**Experimental Designs Or Analyses:**

Yes, seem valid

**Methods And Evaluation Criteria:**

Yes, although the modeling assumption of Eq. (5) seems somewhat arbitrary and there doesn't seem to be clear theoretical reasons presented to back it up (though sec 5.1 gives some empirical evidence)

**Other Comments Or Suggestions:**

NA

**Other Strengths And Weaknesses:**

Strength: The method is based on a natural/simple idea, namely that spoofing attacks generally can only know the red/green tokens for h-grams which they have seen in their "training corpus", and so we can detect spoofing attacks by looking at which h-grams are "spoofed" vs which ones appear in the training corpus.

Weakness: Accordingly, a major weakness of the paper is that (for the (h+1)-gram score method) it requires some good estimate of the "training corpus", which in general is hard (this paper uses C4, which may work for some use cases, but not others).

Weakness: It's not so clear that detecting spoofing attacks is really the right way to go about things: it sort of contributes to a "cat and mouse" game, since one can design spoofing attacks which are harder to detect. The more direct way to go about things seems to be to design watermarking schemes which are harder to spoof.

*** Update: thanks for your responses, which addressed some of my concerns. I have updated my score. ***

**Questions For Authors:**

See above.

**Relation To Broader Scientific Literature:**

One weakness of the paper in relation to the broader scientific literature is that it is not clear that watermarking schemes of many past works are really *intended* to be used for attribution. There are certain papers which specifically study watermarking schmes that are intended to be used for attribution (e.g., http://arxiv.org/abs/2310.18491) in the sense that they should be hard to spoof. Studying spoofing attacks on watermarks which weren't designed to be hard to spoof doesn't seem to be very well-motivated. Accordingly, the paper should make a clearer distinction between watermarking schemes that are designed to be used to publicly attribute text to a model (vs just detecting that the text was produced by the moel -- this distinction is made clear in e.g. https://arxiv.org/abs/2402.09370).

**Theoretical Claims:**

The paper is empirical in nature.

---

> ### Author Rebuttal · Authors · 2025-03-31
>
> $\newcommand{\D}{\mathcal{D}}$$\newcommand{\T}{\tilde{\D}}$We thank the reviewer for their helpful feedback and address their individual questions below.
>
> **Q1: Does the proposed method require a good estimate of the spoofer's training corpus in order to be applicable?**.
>
> A reasonable estimate is enough for the detection to maintain sufficient power. For this, we have already included two experiments in App. D (Figure 8) to ensure that the detection remains accurate despite a weaker estimate of the spoofer’s training corpus.
>
> Let $\D$ be the spoofer's training corpus and $\T$ the estimate of such a corpus. For the first experiment, we build several $\T$ by adding random noise to $\D$. This allows us to control the distance between $\T$ and $\D$ and observe how the power decreases with the distance. For the second experiment, we show that for realistic choices of $\T$ (e.g., C4, Wikipedia), the power of the test remains very similar to the best-case scenario of $\T = \D$. This means that ultimately a reasonable choice of $\T$ is sufficient to maintain high power.
>
> Importantly, as we argue in App. D, the detection can select an appropriate $\T$ from a similar domain as the received text  to be tested. Because the spoofer wants to maximize his chance of successful spoofing, it is reasonable to assume that $\D$ is from the similar domain as the spoofed text he aims to generate, making it likely that the selected $\T$ is close to $\D$.
>
> **Q2: Are the schemes studied by the authors really intended for attribution? Can the authors better motivate the choice of watermarking schemes they study?**
>
> Schemes such as Red-Green, KTH, and AAR have been the focus of multiple previous works considering them for attribution, as shown by spoofing attacks [1,2,3] and multi-bit watermarking [4]. Hence, these schemes have attracted significant attention from researchers exploring broader applications beyond mere detection, including attribution.
>
> Further, such schemes have practical relevance as shown by the deployment of Google Deepmind’s SynthID, the first publicly acknowledged deployment of LLM watermarks. We therefore argue that their prominence and practicality motivate further studies on their real-world security.
>
> Yet, we agree with the reviewer that making a clear distinction as to whether schemes are explicitly designed for attribution is particularly relevant. Based on the reviewer’s suggestion, we will expand our introduction by explicitly discussing watermark attributability by design as well as its current treatment in the research community.
>
> **Q3: Does detecting spoofing attempts lead to a “cat and mouse” game? Isn’t it better to design schemes explicitly designed for attribution?**
>
> We agree with the reviewer that, if the desire is to build an un-spoofable scheme, designing a scheme specifically for attribution, such as [5,6], is the best practice.
>
> Yet, for practical use, practitioners desire multiple, often conflicting, properties: quality, practicality, security, and robustness. Schemes that are harder to spoof might compromise too heavily on these other desirable properties. Importantly, our work indicates that previously successful spoofing attacks on prominent schemes actually are detectable—ultimately making it harder for real-world adversaries to attack such schemes. Further, especially with the increasing work in watermark spoofing on these schemes, we see our work as a helpful contribution to the field by providing general insights into the properties and limitations of learning-based spoofers.
>
> While we ultimately agree that providing an un-spoofable scheme that achieves all of the above properties would be the optimal solution, we see a lot of value in (1) exploring the properties of popular and used watermarks and (2) raising the bar for successful attacks which can yield practical real-world benefits.
>
> **Q4: Can the authors provide more theoretical insights for Eq. (5)?**
>
> Rigorously proving Eq. (5) would require further assumptions about the dependence between $X$ and $Y$. With Lemma 4.1, we assumed independence to justify the asymptotic normality, and with the Unigram score, we provided intuitive reasons why this independence assumption might hold in practice.
>
> For the general case, we show in Appendix C that the independence is violated. While a more general assumption about the dependence structure could allow us to prove Eq. (5), we believe that this complicates the problem modelling without providing significant value to our contribution (as we would still likely rely on empirical evidence to justify our dependence structure assumption). Thus, as noted by the reviewer, we opted for a more direct approach by providing empirical evidence to support Eq. (5).
>
> [1] https://arxiv.org/abs/2402.19361 \
> [2] https://arxiv.org/abs/2312.04469 \
> [3] https://arxiv.org/abs/2405.19677 \
> [4] https://arxiv.org/abs/2308.00221 \
> [5] https://arxiv.org/abs/2402.09370 \
> [6] https://arxiv.org/abs/2310.18491

---

### Official Review · Reviewer_Ai6u · 2025-03-13

**Overall Recommendation:** 4

**Summary:**

This paper investigates whether learning-based attacks that attempt to spoof watermarking schemes in language models leave detectable artifacts in the generated text. The authors propose a statistical method to distinguish genuinely watermarked text from spoofed text by modeling the relationship between the observed watermark color sequence and the frequency of (h+1)-grams in the spoofer’s training data. They introduce a reprompting-based approach that generates new text samples to approximate the expected correlation in genuine watermarked text, enabling reliable detection of spoofed text.

**Claims And Evidence:**

Yes.

**Essential References Not Discussed:**

No.

**Experimental Designs Or Analyses:**

Yes. I checked all parts.

**Methods And Evaluation Criteria:**

Yes.

**Other Comments Or Suggestions:**

No.

**Other Strengths And Weaknesses:**

Pros:

1. This paper is novel and well-organized. It introduces a new statistical approach for detecting spoofed watermarked text, highlighting the challenge of forging watermarks without introducing anomalies.

2. This paper uses well-defined statistical tests to evaluate spoofing artifacts. This provides a formal framework that could be extended to other watermarking and detection settings.

Cons:

1. An important issue is that since the detector receiving the text does not have access to the original prompt, this paper reprompts the model using only a prefix of the received text rather than the original prompt used for generation. This approach may introduce distributional drift, potentially making the reprompted text an imperfect control sample. The extent of this drift in their method is not explicitly analyzed, raising concerns about its impact on detection accuracy. A comparison with prompt-preserving reprompting should be conducted to better assess the validity of the proposed approach and strengthen the paper’s conclusions. Additionally, why not involve using LLM-generated text from a similar domain as the detected text to serve as a baseline directly?

2. The method requires additional text generation to establish a statistical baseline, and it should be very expensive for large-scale detection systems.

**Questions For Authors:**

See Weaknesses.

**Relation To Broader Scientific Literature:**

This paper contributes to the broader scientific literature on watermarking for generative models, adversarial attacks against watermarking, and statistical detection of manipulated text. It builds on prior work in probabilistic watermarking (Kirchenbauer et al.) by demonstrating that learning-based spoofing attacks (Jovanovi´c et al.; Gu et al.) introduce detectable artifacts, aligning with forensic linguistics and stylometry research. The study’s reliance on statistical correlation tests and reprompting-based detection parallels prior work in AI-generated text forensics, reinforcing the challenge of forging watermarks without introducing anomalies.

**Theoretical Claims:**

None. No theoretical claims in this paper.

---

> ### Author Rebuttal · Authors · 2025-03-31
>
> We thank the reviewer for their positive feedback and address their individual questions below. We have attached two additional figures [here](https://drive.google.com/file/d/1-bFmwwrN_kGELtsuhMUTbU4gOBB-UBru/view).
>
> **Q1: Can the authors explicitly analyze the distribution shift between Reprompting with or without the original prompts?**
>
> We thank the reviewer for their suggestion and included a new experiment where we compare Reprompting with and without using the original prompt on Stealing with $h=1$.
>
> We see in the attached Figure 1 that estimating the prompt indeed consistently slightly decreases TPR (at worst a 5% TPR at 1% FPR decrease). This suggests that, for non-adversarial cases, the drift incurred by using the beginning of the received text as a proxy for the prompt has a minimal impact on our detection accuracy.
>
> We will add this experiment to the next revision of our paper.
>
> **Q2: Could the authors use a fixed corpus of LLM generated text from the same domain as the received text instead of Reprompting?**
>
> Intuitively, using a corpus of LLM-generated text whose correlation follows the distribution of Eq. (5) would allow us to estimate the mean and perform the test. However, in practice, it is hard to design a general rule establishing what the mean depends on (e.g., is it the topic of the text, is it the type of words used…).
>
> To strengthen our point, we show in the attached Figure 2 an experiment with Distillation, $h=2$, where we estimate the mean and variance using a $\xi$-watermarked text corpus generated from OpenWebText completions. We then apply Eq. (6), where we replace $S(\omega’)$ with the estimated mean. We see that the FPR is higher than what we would expect, suggesting that the estimated mean does not capture the true mean. This is why we argue that Reprompting, albeit more costly, is more reliable.
>
> We will explicitly include this discussion in the paper.
>
> **Q3: Is the method prohibitively expensive for real world usage?**
>
> We do not think so. As spoofing detection is more targeted than watermark detection—it can be run only when one suspects spoofing—performance issues are, by nature, less critical. We also note that, for some settings ($h=3$), the Unigram score is applicable and doesn’t require additional text generation.
> Yet, we agree that our work is a first step in detecting spoofing attempts on tested schemes, and we believe that making the test more efficient is a worthwhile direction for future work.

---

### Decision · Program_Chairs · 2025-05-01

**Decision:**

Accept (poster)

**Comment:**

This paper tackles the problem of spoofing attacks on sampling‑based watermarks for LLM‑generated text. In this setting,  an adversary forges a valid watermark on arbitrary text. Rather than hardening watermark schemes, the authors take the complementary approach of post‑hoc forensic detection.The paper have observed that, modern spoofers  inevitably learn watermark “red/green” token splits only for n‑grams seen in their limited training corpus, leaving detectable distributional gaps on unseen contexts. With this observation, the authors  formalize a simple hypothesis test based on the correlation between watermark bit patterns and (h+1)‑gram frequencies in the spoofer’s training data, estimating the null (“genuine”) distribution via reprompting from the suspected watermarking model. The authors have shown that across multiple watermark schemes (Red‑Green, KTH, AAR) and spoofing attacks, the detector achieves high true‑positive rates at low false‑positive rates, even when using imperfect corpora or only the text prefix as a reprompt surrogate. All 3 reviewers liked the paper’s post‑hoc detection perspective, its statistical formulation, and its strong empirical performance across varied spoofers. Therefore, it is reasonable to accept the paper as is.